# StreamingBench: Assessing the Gap for MLLMs to Achieve Streaming Video Understanding

## Abstract

The rapid development of Multimodal Large Language Models (MLLMs) has expanded their capabilities from image comprehension to video understanding. However, most of these MLLMs focus primarily on offline video comprehension, necessitating extensive processing of all video frames before any queries can be made. This presents a significant gap compared to the human ability to watch, listen, think, and respond to streaming inputs in real time, highlighting the limitations of current MLLMs. In this paper, we introduce **StreamingBench**, the first comprehensive benchmark designed to evaluate the streaming video understanding capabilities of MLLMs. StreamingBench assesses three core aspects of streaming video understanding: (1) **real-time visual understanding**, (2) **omni-source understanding**, and (3) **contextual understanding**. The benchmark consists of 18 tasks, featuring 900 videos and 4,300 human-curated QA pairs. Each video features five questions presented at different time points to simulate a continuous streaming scenario. We conduct experiments on StreamingBench with 13 open-source and proprietary MLLMs and find that even the most advanced proprietary MLLMs like Gemini 1.5 Pro and GPT-4o perform significantly below human-level streaming video understanding capabilities. We hope our work can facilitate further advancements for MLLMs, empowering them to approach human-level video comprehension and interaction in more realistic scenarios.

## 1 Introduction

The rapid evolution of Multimodal Large Language Models (MLLMs) has significantly reshaped the field of Artificial Intelligence (Yang et al., 2023; Reid et al., 2024; Liu et al., 2024c;a). Current advanced MLLMs (Reid et al., 2024; Wang et al., 2024a; Yao et al., 2024) have already demonstrated exceptional performance in video understanding tasks, excelling on existing video benchmarks (Fu et al., 2024; Wang et al., 2024b; Zhou et al., 2024; Ataallah et al., 2024). Moreover, several pioneering studies (Chen et al., 2024a; Zhang et al., 2024a; Wu et al., 2024) have focused on improving the ability of MLLMs to comprehend real-time online video streams, pushing the boundaries of their applicability and efficiency in dynamic environments. In the industry, streaming video understanding has also attracted significant attention, with OpenAI's GPT-4o (OpenAI, 2024) as a prominent example that demonstrates human-like perception and understanding of streaming inputs.

Despite the recognition of the importance of streaming video understanding for MLLMs, most existing video understanding benchmarks (Fu et al., 2024; Wang et al., 2024b; Zhou et al., 2024) are primarily designed for offline evaluation. In such setups, all video frames are pre-loaded into the MLLMs before any queries are made, assuming the model has complete access to the entire video content. In contrast, streaming video understanding tasks differ in three key aspects: (1) queries can arise at any point during the video stream, rather than just at the end; (2) synchronized visual and audio inputs must be considered as in real-world streaming scenarios; (3) the influence of context must be taken into account, such as redundant information in long video streams and the history of streaming interactions. These differences in design principles between offline and streaming tasks make it quite challenging to adapt offline benchmarks for streaming evaluation.

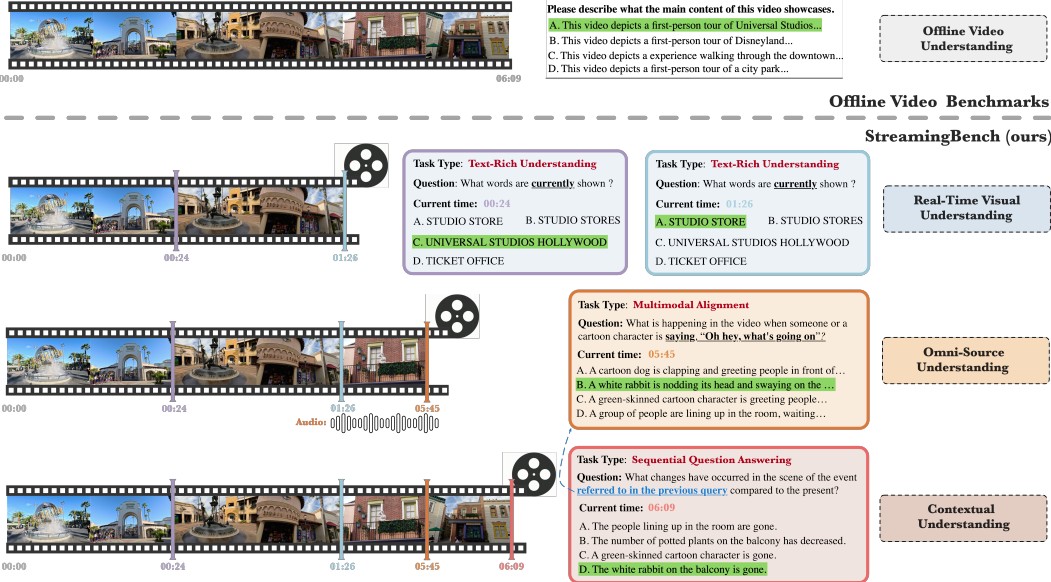

Figure 1: Illustrative comparison between StreamingBench and previous offline video benchmarks. In offline video benchmarks, questions are designed based on the entire video being visible. In contrast, StreamingBench presents questions at specific moments, with three main task categories specifically designed to evaluate fundamental capabilities in streaming video understanding.

To the best of our knowledge, the only current benchmark related to streaming video understanding is VStream-QA (Zhang et al., 2024a). The main attribute of VStream-QA is that each question-answer pair is assigned a timestamp indicating its position in the video and is only related to the content preceding that point. However, VStream-QA includes only 32 videos from Ego4d (Grauman et al., 2022) and MovieNet (Huang et al., 2020), with a limited variety of video types and a narrow range of scenarios. In addition, it only covers five types of tasks, focuses solely on the visual modality, and the questions for each video are independent of each other. These limitations prevent VStream-QA from fully assessing streaming video understanding abilities for MLLMs when confronted with complex, multimodal streaming inputs in real-world scenarios.

To address the limitations of existing video benchmarks, we introduce **StreamingBench**, the first comprehensive benchmark for assessing the streaming video understanding capabilities of MLLMs. StreamingBench consists of 900 videos and 4,300 questions, spanning eight diverse video categories that reflect a wide range of real-world scenarios. Each video features five questions that are manually curated to ensure a high level of relevance to the streaming video scenarios. These questions are categorized into 18 tasks, and based on the characteristics of streaming video tasks, they can be grouped into three main categories as illustrated in Figure 1:

- **Real-Time Visual Understanding**, which focuses on the ability of MLLMs to comprehend visual content in real-time, recognizing and interpreting objects, actions, and changes as they happen within the video stream. For example, in Figure 1, the answer to the question "*What words are currently shown?*" may vary depending on the specific moment in time the question is asked, highlighting the dynamic nature of streaming video tasks.
- **Omni-Source Understanding**, which refers to the ability integrate visual and audio information in real-time video streams. MLLMs must handle both sources simultaneously to provide a comprehensive understanding of the scene and answer questions that depend on their synchronization, such as "*What is happening in the video when [sound] is made?*".
- **Contextual Understanding**, which evaluates the capability of MLLMs to comprehend the broader context within a video stream, including detecting anomalies, filtering misleading information, maintaining continuity across sequential interactions, and responding proactively based on predefined conditions. For instance, as shown in the last query of Figure 1, a follow-up question is asked based on the content of the previous query interaction, with a reference to "*the event referred to in the previous query*".

Table 1: Comparison between StreamingBench and other video benchmarks. Timestamp denotes whether to assign timestamps to questions. Temporal Clues denote whether the questions are related to different temporal clues within videos (Section 4.4)). SQA and PO denote sequential question answering and proactive output, respectively (Section 3.1.3).

| | Benchmark | #Videos | #QA Pairs | Timestamp | Temporal Clues | | | Modality | | Streaming Interaction | | Annotation | |
|---|---|---|---|---|---|---|---|---|---|---|---|---|---|
| | | | | | Prior | Concurrent | Subsequent | Vision | Audio | SQA | PO | Auto | Manual |
| Offline (Short) | MSRVTT-QA (Xu et al., 2017) | 2,990 | 72,821 | ✗ | ✓ | ✗ | ✗ | ✓ | ✗ | ✗ | ✗ | ✓ | ✗ |
| | TGIF-QA (Jang et al., 2017) | 9,575 | 8,506 | ✗ | ✓ | ✗ | ✗ | ✓ | ✗ | ✗ | ✗ | ✓ | ✓ |
| | MV-Bench (Li et al., 2024b) | 3,641 | 4,000 | ✗ | ✓ | ✗ | ✗ | ✓ | ✗ | ✗ | ✗ | ✓ | ✗ |
| | How2QA (Li et al., 2020) | 1,166 | 2,852 | ✗ | ✓ | ✗ | ✗ | ✓ | ✗ | ✗ | ✗ | ✗ | ✓ |
| | ActivityNet-QA (Yu et al., 2019) | 800 | 8,000 | ✗ | ✓ | ✗ | ✗ | ✓ | ✗ | ✗ | ✗ | ✗ | ✓ |
| Offline (Long) | InfiniBench (Ataallah et al., 2024) | 1219 | 108,200 | ✗ | ✓ | ✗ | ✗ | ✓ | ✗ | ✗ | ✗ | ✓ | ✓ |
| | MLVU (Zhou et al., 2024) | 1,334 | 2,593 | ✗ | ✓ | ✗ | ✗ | ✓ | ✗ | ✗ | ✗ | ✓ | ✓ |
| | LVBench (Wang et al., 2024b) | 500 | 1,549 | ✗ | ✓ | ✗ | ✗ | ✓ | ✗ | ✗ | ✗ | ✗ | ✓ |
| | Video-MME (Fu et al., 2024) | 900 | 2,700 | ✗ | ✓ | ✗ | ✗ | ✓ | ✓ | ✗ | ✗ | ✗ | ✓ |
| Online | VStream-QA (Zhang et al., 2024a) | 32 | 3,500 | ✓ | ✓ | ✓ | ✗ | ✓ | ✗ | ✗ | ✗ | ✓ | ✗ |
| | **StreamingBench(Ours)** | 900 | 4,300 | ✓ | ✓ | ✓ | ✓ | ✓ | ✓ | ✓ | ✓ | ✓ | ✓ |

We conduct experiments on StreamingBench with state-of-the-art MLLMs, including three proprietary models GPT-4o (OpenAI, 2024), Gemini 1.5 Pro (Reid et al., 2024) and Claude 3.5 Sonnet (Anthropic, 2024), and 10 advanced open-source MLLMs like LLaVA-OneVision (Li et al., 2024a), Qwen2-VL (Wang et al., 2024a) and MiniCPM-V 2.6 (Yao et al., 2024). Since these models currently cannot accept streaming video input[1], we convert each streaming task into an offline one for evaluation. For each question, the model processes the video segment from the start to the point when the question is asked, treating it as the complete input, and provides a response based on that segment. The results show that even the best-performing model, Gemini 1.5 Pro, achieves only an average accuracy of 67.36%, which is 24.30% lower than human performance. This indicates that there is a significant gap between MLLMs and human performance in understanding video streams.

To further investigate this gap, we conduct a series of analytical experiments, revealing that current models perform poorly in terms of real-time processing. This may be attributed to the fact that most existing MLLMs are primarily trained on offline videos. Additionally, we find that these models generally lack the ability to understand and interact with streaming contexts. Specifically, redundant information in the context of streaming videos significantly affects model performance, and current models struggle with proactive output in streaming scenarios and fail to effectively respond to continuous queries. We hope these findings will provide valuable insights for improving future MLLMs and contribute to the development of the next generation of multimodal systems.

## 2 RELATED WORK

**Video MLLMs.** Recently, the development of advanced MLLMs has shifted from single image understanding to video comprehension (Reid et al., 2024; Wang et al., 2024a; Yao et al., 2024; Lin et al., 2023; Chen et al., 2024b; Li et al., 2024a). These video MLLMs typically work by converting entire videos into visual tokens that can be processed by LLMs, through sampling and encoding video frames. However, these models are limited to offline video understanding rather than real-time, real-world streaming video comprehension. In contrast, GPT-4o (OpenAI, 2024) explores the potential for human-like perception and understanding of streaming inputs. There are also several streaming video MLLMs in the academic field, including VideoLLM-online (Chen et al., 2024a), Flash-VStream (Zhang et al., 2024a), and VideoLLM-MoD (Wu et al., 2024). With the growing interest in research on streaming video MLLMs, there is an increasing urgency to comprehensively evaluate their streaming video understanding capabilities.

**Video Understanding Benchmarks.** The development of video understanding benchmarks has progressed in tandem with advancements in MLLMs. Most current benchmarks are primarily focused on evaluating capabilities of either comprehensive video understanding (Li et al., 2024b; Fu et al., 2024) or long-form video understanding (Wang et al., 2024b; Zhou et al., 2024). To our knowledge, there is currently only one benchmark, VStream-QA (Zhang et al., 2024a), that is related to streaming video understanding, where each question is assigned a timestamp to simulate a real-time

---

[1]The GPT-4o API currently does not support video inputs.

query. However, VStream-QA has limitations in terms of the video types and task designs it encompasses, making it not suitable for a thorough evaluation of streaming video understanding abilities. In this paper, we introduce StreamingBench, a comprehensive streaming understanding benchmark. A comparison between StreamingBench and other video benchmarks is provided in Table 1.

# 3 STREAMINGBENCH

## 3.1 TAXONOMY

We identify three key distinctions between a streaming video understanding benchmark and traditional offline video benchmarks: (1) the inclusion of real-time queries that can appear at any point during the video stream, rather than solely at the end; (2) the consideration of synchronized visual and audio content, mirroring real-world video streams; and (3) the reflection of the complex and dynamic context of video streams, encompassing the evaluation of streaming interactions beyond conventional isolated question answering. Based on these distinctions, we design three task categories: **Real-Time Visual Understanding**, **Omni-Source Understanding** and **Contextual Understanding**. Each category mainly addresses one of these distinctions and evaluates specific core capabilities essential for streaming video comprehension.

### 3.1.1 REAL-TIME VISUAL UNDERSTANDING

The tasks in this category aim to assess the ability of a model to perceive, comprehend, and reason based on the visual content of video streams. Each question is accompanied by a timestamp that indicates the time of the query and ensures that it only pertains to the visual content preceding that specific moment. To emphasize the real-time nature of the questions, they include specific time indicators such as "right now", "just now", or "currently". As a result, the same question asked at different times may yield different answers.

There are 10 tasks that belong to this category: (1) **Object Perception (OP)**: Detect and identify specific objects within the video. (2) **Causal Reasoning (CR)**: Analyze cause-and-effect relationships in events. (3) **Clips Summarization (CS)**: Summarize main content in specific video clips. (4) **Attribute Perception (ATP)**: Identify and categorize object or individual attributes. (5) **Event Understanding (EU)**: Recognize and describe sequences of events. (6) **Text-Rich Understanding (TR)**: Interpret and explain text-rich content within the video. (7) **Prospective Reasoning (PR)**: Predict future events based on current video context. (8) **Spatial Understanding (SU)**: Understand and describe spatial relationships and locations. (9) **Action Perception (ACP)**: Identify and describe specific actions in the video. (10) **Counting (CT)**: Count occurrences of specific objects or actions. These tasks cover the main visual understanding tasks and effectively evaluate the ability of MLLMs to understand visual information in real-time in streaming scenarios. For deterministic evaluations, all test examples are presented as multiple-choice questions with four distinct options each. For examples of each task, please refer to Appendix D.

### 3.1.2 OMNI-SOURCE UNDERSTANDING

These tasks evaluate the capability of a model to process visual and audio content in a video stream simultaneously, especially focusing on the ability to integrate information from video and audio content, or align them temporally. There are four tasks in this category:

**Emotion Recognition (ER):** *What is the mood of the person?* The task involves identifying the current emotion of a particular person in the video and determining the cause of their emotional change, based on the visual and auditory cues in the video stream.

**Scene Understanding (SCU):** *Describe the scene that just occured.* This task requires MLLMs to comprehend and describe dynamic scenes as they occur in a video stream, with a specific emphasis on accurately identifying both the visual elements and the audio that occurs simultaneously.

**Source Discrimination (SD):** *Who just said "[quote]"?* This task requires MLLMs to accurately identify the speaker of specific lines of dialogue ([quote]) within a video stream, based on the visual and auditory cues presented just before or during the time the dialogue was spoken.

**Multimodal Alignment (MA):** *Describe the scene when a person said "[quote]".* This task requires MLLMs to accurately correlate spoken words ([quote]) with corresponding visual scenes in a video. Based on the time intervals and context provided, MLLMs must describe the scene that occurs when a specific line is spoken, ensuring that the visual and auditory elements are correctly aligned.

As with the questions in the previous task type, each question in omni-source understanding is set with a specific timestamp, and is a multiple-choice question with four options for the purpose of deterministic evaluation. In addition, we make sure that all questions can not be answered without understanding both visual and audio content. Please refer to Appendix D for data examples.

### 3.1.3 CONTEXTUAL UNDERSTANDING

These tasks focus on assessing the ability of MLLMs to provide accurate responses based on complex context within a continuous video stream. Such context includes not only the redundant information presented throughout the video, but also the the streaming interactions such as prior question-answer pairs or conditions for late proactive outputs. Overall, there are four contextual understanding tasks. The first two involve filtering information from the redundant context:

**Misleading Context Understanding (MCU):** *What are the cards on the table right now?* In video streams, misleading context can lead models to make false predictions. For instance, when playing cards, different cards may have appeared on the table in previous video frames. To answer this example question, the model must distinguishing the current state of the cards from that appeared in earlier frames but are no longer present. This task challenges the model to maintain precision in scenarios where similar but incorrect visual cues are prevalent, ensuring reliable understanding in complex visual environments.

**Anomaly Context Understanding (ACU):** *What unusual event just occurred?* This task evaluate the MLLMs' ability to detect and accurately identify unusual or unexpected events within a video stream. The model must differentiate between subtle variations in similar scenarios and correctly identify the anomaly, ensuring precise understanding in dynamic and unpredictable environments.

The form of these two tasks is the same as previous questions, i.e., multi-choice questions with assigned timestamps. There are also two tasks related to *streaming interactions*:

**Sequential Question Answering (SQA):** *What is the current outfit of the person mentioned in the first question?* This task is characterized by a sequence of questions where each subsequent question is directly related to the entity or event mentioned in previous ones. The model must effectively utilize episodic memory to accurately link related information, ensuring coherent and contextually relevant responses throughout the task sequence.

**Proactive Output (PO):** *When a goal is scored, output "GOAL".* Unlike typical input-output tasks where the model responds directly to the input, this task requires the model to proactively determine when to generate output based on predefined conditions. This involves maintaining an internal state to track relevant information from incoming video frames, which is crucial for responsive AI systems in real-time streaming environments.

The question format of SQA is similar to other formats but includes an additional history of QA sequences. In contrast, each question in the PO task includes an additional timestamp, indicating the exact time when the output should occur. Data examples are in Appendix D.

### 3.2 DATA CONSTRUCTION

**Video Selection.** We divide the streaming understanding scenarios into eight distinct categories to ensure a comprehensive simulation of real-world, real-time streaming applications: *life record*, *competition*, *education*, *TV show*, *video games*, *documentary*, *animation & movie* and *unusual event*. We manually select and carefully curate 900 YouTube videos to cover all of these scenarios and ensure that they possess attributes suited for different streaming video understanding tasks.

**QA Generation.** We use a hybrid annotation pipeline to generate QA pairs for different task categories in StreamingBench. For real-time visual understanding tasks and the proactive output task, we first sample frames from the video at 1 fps and use GPT-4o to generate captions for every 20

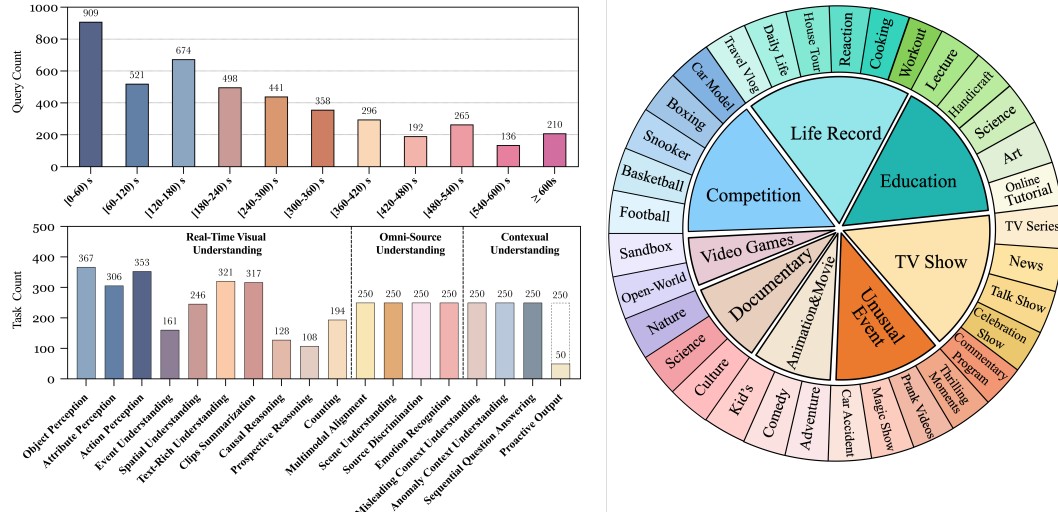

Figure 2: The **left** diagrams depict the distribution of tasks and video durations in StreamingBench, while the **right** diagram illustrates the categories of the 900 videos included in the benchmark. It is important to note that we have created a total of 250 questions for the proactive output task, but for efficiency, only 50 of them are currently evaluated in the present version of StreamingBench. We plan to release the remaining questions to support future evaluations.

frames. Since StreamingBench requires queries at various points in the video, we add a timestamp to the top-left corner of each frame, which allows GPT-4o to create captions with finer temporal granularity (with intervals of less than 20 seconds). Using these timestamped, fine-grained captions, GPT-4o then generates QA pairs for different tasks and automatically assign a question timestamp. For omni-source understanding tasks and other contextual understanding tasks, we ask human annotators to manually label the QA pairs.

**Quality Control.** To ensure the quality of data in StreamingBench, we implement a rigorous human verification process for both automatically generated and manually annotated QA pairs. Each pair is reviewed for accuracy, clarity, and relevance. Low-quality pairs containing ambiguities or incorrect labels are revised, and questions that can be answered without video information are discarded. Additionally, we shuffle options to ensure a balanced distribution. This meticulous quality control process ensures that StreamingBench effectively challenges models to demonstrate their real-time streaming video understanding capabilities. More details of data construction are in Appendix 3.2.

Figure 2 depicts the main statistics of StreamingBench, which comprises 900 videos and 4,300 questions. The videos span eight different categories, with durations ranging from as short as 3 seconds to as long as 24 minutes, covering a wide range of real-world streaming scenarios. Specifically, the real-time visual understanding category includes 500 videos with a total of 2,500 questions, the omni-source understanding category comprises 200 videos with 1,000 questions, and the contextual understanding category contains 200 videos with 800 questions.

## 4 EXPERIMENTS

In this section, we present the experimental setup, evaluation results, and analysis of Streaming-Bench. We evaluate 13 open-source and proprietary MLLMs, highlighting their strengths and limitations in streaming video scenarios. Building on these initial findings, we then conduct additional in-depth analytical experiments to further explore their performance, aiming to facilitate further advancements for MLLMs in enhancing its streaming video understanding capabilities.

### 4.1 SETTINGS

We evaluate three proprietary MLLMs: GPT-4o (OpenAI, 2024), Gemini 1.5 Pro (Reid et al., 2024), and Claude 3.5 Sonnet (Anthropic, 2024), alongside 10 high-performing open-source video

Table 2: Performance of various MLLMs on StreamingBench. †: For videos of varying lengths, we apply the corresponding frame rates for Qwen2-VL: 1 fps for under 5 minutes, 0.5 fps for 5 to 10 minutes, and 0.2 fps for over 10 minutes, balancing efficiency and visual information retention. ‡: Human evaluation with a randomly sampled 10% of all questions, as detailed in Appendix C.2.

| Model | Params | Frames | Real-Time Visual Understanding | | | | | | | | | | | Omni-Source Understanding | | | | | Contextual Understanding | | | | | Overall |
|---|---|---|---|---|---|---|---|---|---|---|---|---|---|---|---|---|---|---|---|---|---|---|---|---|
| | | | OP | CR | CS | ATP | EU | TR | PR | SU | ACP | CT | All | ER | SCU | SD | MA | All | ACU | MCU | SQA | PO | All | |
| **Human** | | | | | | | | | | | | | | | | | | | | | | | | |
| Human‡ | - | - | 89.47 | 92.00 | 93.60 | 91.47 | 95.65 | 92.52 | 88.00 | 88.75 | 89.74 | 91.30 | 91.46 | 88.00 | 88.24 | 93.60 | 90.27 | 90.26 | 88.80 | 90.40 | 95.00 | 100 | 93.55 | 91.66 |
| **Proprietary MLLMs** | | | | | | | | | | | | | | | | | | | | | | | | |
| Gemini 1.5 pro | - | 1 fps | 79.02 | **80.47** | 83.54 | 79.67 | **80.00** | **84.74** | **77.78** | 64.23 | **71.95** | 48.70 | **75.69** | **46.80** | **39.60** | **74.90** | **80.00** | **60.22** | **51.41** | **40.73** | 54.80 | 30.00 | **47.79** | **66.90** |
| GPT-4o | - | 64 | 77.11 | **80.47** | **83.91** | 76.47 | 70.19 | 83.80 | 66.67 | 62.19 | 69.12 | **49.22** | 73.28 | 41.20 | 37.20 | 43.60 | 56.00 | 44.50 | 41.20 | 38.40 | 32.80 | 29.41 | 36.96 | 59.83 |
| Claude 3.5 Sonnet | - | 20 | **80.49** | 77.34 | 82.02 | **81.73** | 72.33 | 75.39 | 61.11 | 61.79 | 69.32 | 43.09 | 72.44 | 31.60 | 34.00 | 32.80 | 48.80 | 36.80 | 38.40 | 34.80 | 34.40 | **35.29** | 35.83 | 57.34 |
| **Open-Source Video MLLMs** | | | | | | | | | | | | | | | | | | | | | | | | |
| LLaVA-OneVision | 7B | 32 | **80.38** | 74.22 | 76.03 | **80.72** | **72.67** | **71.65** | 67.59 | **65.45** | **65.72** | 45.08 | **71.12** | 40.80 | **37.20** | 33.60 | **44.80** | **38.40** | **35.60** | **36.00** | 27.27 | 11.76 | 31.63 | **56.16** |
| Qwen2-VL | 7B | 0.2-1 fps† | 75.20 | 82.81 | 73.19 | 77.45 | 68.32 | 71.03 | **72.22** | 61.19 | 61.47 | 46.11 | 69.04 | 41.20 | 22.00 | 32.80 | 43.60 | 34.90 | 31.20 | 26.00 | 39.60 | 1.96 | 30.37 | 53.91 |
| MiniCPM-V 2.6 | 8B | 32 | 71.93 | 71.09 | **77.92** | 75.82 | 64.60 | 65.73 | 70.37 | 56.10 | 62.32 | 53.37 | 67.44 | 40.80 | 24.00 | 34.00 | 41.20 | 35.00 | 34.00 | 31.60 | **41.92** | 9.80 | **34.21** | 53.71 |
| LLaVA-NeXT-Video | 32B | 64 | 78.20 | 70.31 | 73.82 | 76.80 | 63.35 | 69.78 | 57.41 | 56.10 | 64.31 | 38.86 | 66.96 | 37.69 | 24.80 | 34.40 | 42.80 | 34.90 | 29.20 | 30.40 | 35.35 | 5.88 | 30.04 | 52.64 |
| InternVL-V2 | 8B | 16 | 68.12 | 60.94 | 69.40 | 77.12 | 67.70 | 62.93 | 59.26 | 53.25 | 54.96 | **56.48** | 63.72 | 37.60 | 26.40 | **37.20** | 42.00 | 35.80 | 32.00 | 31.20 | 32.32 | 11.76 | 30.59 | 51.06 |
| Kangaroo | 7B | 64 | 71.12 | **84.38** | 70.66 | 73.20 | 67.08 | 61.68 | 56.48 | 55.69 | 62.04 | 38.86 | 64.60 | 37.60 | 31.20 | 28.80 | 39.20 | 34.20 | 32.80 | 26.40 | 33.84 | 3.92 | 29.32 | 50.97 |
| LongVA | 7B | 128 | 70.03 | 63.28 | 61.20 | 70.92 | 62.73 | 59.50 | 61.11 | 53.66 | 54.67 | 34.72 | 59.96 | 39.60 | 32.40 | 28.00 | 41.60 | 35.40 | 32.80 | 29.60 | 30.30 | 5.88 | 29.34 | 48.55 |
| VILA-1.5 | 8B | 14 | 53.68 | 49.22 | 70.98 | 56.86 | 53.42 | 53.89 | 54.63 | 48.78 | 50.14 | 17.62 | 52.32 | **41.60** | 26.40 | 28.40 | 36.00 | 33.10 | 26.80 | 34.00 | 23.23 | **15.68** | 27.24 | 43.18 |
| Video-CCAM | 14B | 96 | 56.40 | 57.81 | 65.30 | 62.75 | 64.60 | 51.40 | 42.59 | 47.97 | 49.58 | 31.61 | 53.96 | 33.60 | 22.00 | 28.40 | 34.80 | 29.70 | 27.60 | 24.40 | 16.67 | 5.88 | 21.83 | 42.34 |
| Video-LLaMA2 | 7B | 32 | 55.86 | 55.47 | 57.41 | 58.17 | 52.80 | 43.61 | 39.81 | 42.68 | 45.61 | 35.23 | 49.52 | 30.40 | 32.40 | 30.40 | 36.00 | 32.40 | 24.80 | 26.80 | 18.67 | 1.96 | 22.08 | 40.43 |

MLLMs: Video-LLaMA2 (Zhang et al., 2023), MiniCPM-V 2.6 (Yao et al., 2024), InternVL-V2 (Chen et al., 2024c), Video-CCAM (Fei et al., 2024), LongVA (Zhang et al., 2024b), LLaVA-OneVision (Li et al., 2024a), VILA-1.5 (Fang et al., 2024), Kangaroo (Liu et al., 2024d), LLaVA-NeXT-Video (Liu et al., 2024b), and Qwen2-VL (Wang et al., 2024a).[2] We adhere to the official configurations of most MLLMs for frame extraction from the videos, as detailed in Appendix A.1.

Since current MLLMs lack the ability to accept streaming video input, we convert each streaming task into an offline task for evaluation except for the proactive output task. During the evaluation process, each video is clipped into the segment from the beginning to the timestamp when the question is asked. Then the model answers the question based on this video segment in an offline way. We use accuracy as the evaluation metric for all multiple-choice questions.

For SQA, the basic evaluation process and metric are consistent with other tasks. The only difference is that contextual information, i.e., previous QA pairs should be additionally included. For simplicity, we attach the history of question-answer pairs before the current question to expand the input as: "{Timestamp1}: {QA1} . . .; Answer the question accordingly: {current question}".

For the Proactive Output task, most models cannot be directly evaluated, as they lack the ability to autonomously provide output without prompts. To address this, we implement a polling strategy: we define an interval spanning several seconds before and after the ground truth timestamp (the moment when the model is expected to output). During this interval, the model is queried every second with the question "Is it the right time to output?" This continues until the model responds with "Yes." At that point, the model is prompted to provide the relevant keywords, and this moment is recorded as the actual output timestamp. A question in the PO task is considered accurately resolved only if the difference between the actual output timestamp and the ground truth timestamp is less than two seconds. The average accuracy across all queries is then computed and used as the performance metric for the PO task. Please refer to Appendix A.2 for more evaluation protocols.

## 4.2 RESULTS ON STREAMINGBENCH

The performance of 13 open-source and proprietary models on the 18 tasks of StreamingBench is presented in Table 2. The results indicate that all three proprietary models outperform the best-performing open-source model, LLaVA-OneVision, with Gemini 1.5 pro achieving the highest score of 67.36%. Among the open-source models, LLaVA-OneVision ranks first with a score of 54.79%, followed closely by Qwen2-VL and MiniCPM-V 2.6, which achieve scores of 52.69% and 52.58%, respectively. For comparison, we sample 10% of the tasks from each of the 18 tasks for human

---

[2]We also evaluate two streaming video MLLMs claiming online processing capabilities: VideoLLM-Online (Chen et al., 2024a) and Flash-VStream (Zhang et al., 2024a). However, the performance of these two models is relatively poor. We list the evaluation results of them in Appendix C.1.

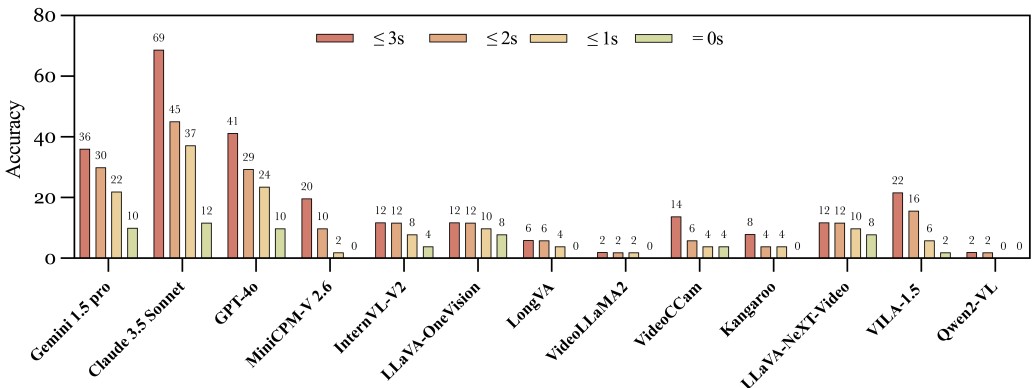

Figure 3: Performance of different MLLMs on the proactive output task. "$\leq x$s" means that the answer is considered correct if the actual output time is within $x$ seconds of the ground truth.

evaluation. The average human score across 18 tasks is 91.66%. Even the best-performing MLLMs, Gemini 1.5 Pro, lags significantly behind human performance.

The results demonstrate that all models perform well on real-time visual understanding tasks, but exhibit generally poor performance on omni-source understanding and contextual understanding tasks. This suggests that the models' ability to understand offline video transfers effectively to real-time visual understanding, but they struggle with tasks that require audio information for omni-source understanding and tasks that demand consideration of contextual information in scenarios with streaming interactions or high-redundancy visual inputs for contextual understanding. This highlights a significant gap between the current MLLMs and the goal of achieving streaming video understanding. Notably, Gemini 1.5 Pro excels in omni-source understanding due to its capability to process audio within videos. Additionally, Claude 3.5 Sonnet achieves the highest score among all models in the proactive output task, with a score of 45.10% within a two-second error margin. The decent performance of these proprietary models on omni-source understanding and contextual understanding tasks reflects the potential of these models to achieve streaming video understanding.

## 4.3 MODEL PERFORMANCE ON DIFFERENT VIDEO LENGTHS

We further investigate the impact of video length on the model capabilities of streaming video understanding. As most current MLLMs can process minute-level videos, we choose 60 seconds as a threshold to distinguish between short and long videos, and compare the models' performance on both. We focus on the top three open-source models with the highest performance in real-time visual understanding. The results, as shown in Table 3, indicate that all models perform worse overall on videos longer than 60 seconds compared to their performance on shorter videos. However, Qwen2-VL stands out by demonstrating better performance on long videos than short ones in the tasks of Causal Reasoning (CR) and Clip Summarization (CS). This highlights the need for improvements in the ability of MLLMs to effectively process longer video content.

Table 3: Performance of the top open-source models on different tasks for videos ≤60s and >60s.

| Model | Video Length | Real-Time Visual Understanding | | | | | | | | | | |
|---|---|---|---|---|---|---|---|---|---|---|---|---|
| | | OP | CR | CS | ATP | EU | TR | PR | SU | ACP | CT | **All** |
| **LLaVA-OneVision** | ≤60 s | 84.81 | 75.00 | 84.93 | 91.30 | 89.29 | 85.88 | 82.61 | 73.91 | 73.53 | 63.26 | 81.30 |
| | >60 s | 79.17 | 74.07 | 72.95 | 76.79 | 66.92 | 66.53 | 63.53 | 63.00 | 63.86 | 25.00 | 66.94 |
| **Qwen2-VL** | ≤60 s | 86.08 | 80.00 | 78.08 | 85.51 | 89.28 | 82.35 | 78.26 | 73.91 | 67.65 | 67.35 | 78.89 |
| | >60 s | 72.22 | 81.18 | 91.30 | 75.11 | 63.91 | 66.95 | 70.59 | 59.50 | 60.00 | 38.89 | 66.33 |
| **MiniCPM-V 2.6** | ≤60 s | 88.61 | 75.00 | 83.56 | 89.86 | 75.00 | 81.18 | 82.61 | 69.57 | 77.94 | 79.59 | 81.67 |
| | >60 s | 67.36 | 70.37 | 76.23 | 71.73 | 62.41 | 60.17 | 67.06 | 53.00 | 58.60 | 44.44 | 63.52 |

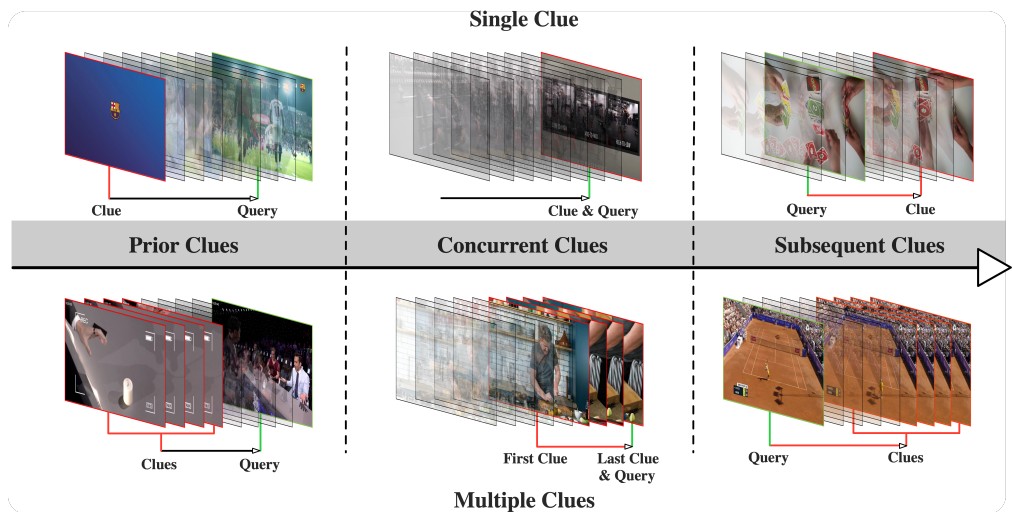

Figure 4: A two-dimensional classification of clues in StreamingBench. The first dimension categorizes clues by their timing relative to the query: *Prior* (before the query), *Concurrent* (during the query), and *Subsequent* (after the query). The second dimension differentiates between *Single Clue*, requiring only one frame, and *Multiple Clues*, needing multiple frames for the answer.

Table 4: The average accuracy of MLLMs on tasks with different clue types.

| Clue Type | Prior | | Concurrent | | Subsequent | | Total | |
|---|---|---|---|---|---|---|---|---|
| | Num. | Acc. | Num. | Acc. | Num. | Acc. | Num. | Acc. |
| **Single** | 212 | 53.91% | 1278 | 43.79% | 32 | 8.93% | 1522 | 44.47% |
| **Multiple** | 1196 | 53.57% | 1564 | 44.07% | 18 | 3.01% | 2778 | 47.83% |
| **Total** | 1408 | 53.75% | 2842 | 43.92% | 50 | 6.72% | 4300 | 46.64% |

## 4.4 MODEL PERFORMANCE ON TASKS WITH DIFFERENT TEMPORAL CLUES

We classify questions according to clue types demonstrated in Figure 4, and show average accuracy of different models in Table 4. The results demonstrate that model performance is not related to the number of clues but rather to the position of clue occurrence. Specifically, models perform better on prior-type tasks than on concurrent- and subsequent-type tasks. This discrepancy is likely due to the fact that most offline video QA tasks in current training datasets focus on prior-type tasks, while concurrent- and subsequent-type tasks are underrepresented. Enhancing the ability of MLLMs to handle concurrent- and subsequent-type tasks is crucial for future progress.

## 4.5 ANALYSES ON CONTEXUAL UNDERSTANDING TASKS

**Do Redundant Information Affect Contextual Understanding?** We observe that all models perform unsatisfactorily in two contextual understanding tasks involving redundant information: misleading and anomaly context understanding tasks (MCU and ACU). To quantitatively assess the impact of highly redundant visual information on model performance, we sample 125 questions from these tasks and manually eliminate redundant information in them. For MCU, we extract frames that contain clues for answering the question and discard other misleading frames. For ACU, we keep only the video segments where the anomaly events occur as inputs. We conduct experiments using four top-performing open-source MLLMs on StreamingBench. The results, as shown in Table 5, indicate that MLLMs consistently achieve better performance when redundant visual information is removed from the inputs. This finding underscores the insufficient robustness of current MLLMs in handling redundant information. Future models should aim to improve their ability to accurately extract relevant information from inputs with high visual redundancy.

Table 5: Comparison of the performance of four models on MCU and ACU tasks with and without high-**R**edundancy visual **I**nformation inputs (**RI**). $\Delta$ denotes the performance difference.

|  | LLaVA-NeXT-Video | | | MiniCPM-V 2.6 | | | Qwen2-VL | | | LLaVA-OneVision | | |
|---|---|---|---|---|---|---|---|---|---|---|---|---|
|  | w/ RI | w/o RI | $\Delta$ | w/ RI | w/o RI | $\Delta$ | w/ RI | w/o RI | $\Delta$ | w/ RI | w/o RI | $\Delta$ |
| **MCU** | 30.40 | 65.60 | **+35.20** | 31.60 | 49.60 | **+18.00** | 26.00 | 67.20 | **+41.20** | 36.00 | 68.00 | **+32.00** |
| **ACU** | 29.20 | 48.00 | **+18.80** | 34.00 | 50.40 | **+16.40** | 31.20 | 53.60 | **+22.40** | 35.60 | 49.60 | **+14.00** |

**Do Question References Constrain Model Performance in Sequential QA?** To understand the impact of references between questions on model performance, we explicitly resolve these references in the original questions and conduct experiments. For example, the original question "How many game scores has the team referred to in the previous question scored so far?" is modified to "How many game scores has GS (*the team name*) scored so far?". (See Figure 14) As shown in the left part of Figure 5, the results indicate that most models, except for MiniCPM-V 2.6, exhibit performance improvement to some extent. This suggests that the suboptimal performance of the models is partly due to their inability to resolve references between questions, requiring further adaptation to the sequential question-answering scenario for MLLMs.

**Why Cannot MLLMs Handle the Proactive Output Task?** We assume that the proactive output (PO) task requires two abilities of an MLLM: (1) accurately locating and responding to critical information in continuously incoming frames, and (2) following proactive output instructions. Based on these two aspects, we further analyze why MLLMs struggle to handle the PO task effectively. First, we relax the evaluation threshold for the time difference between the actual output time and the ground truth timestamp, and observe a rapid improvement in accuracy as shown in Figure 3. This suggests that MLLMs have a certain ability to respond to information, but lack precision in timing. Next, we transform the PO task into a more traditional "passive" output task, where we directly query the model for critical information at the ground truth timestamp and assess the correctness of the response. For example, the original question "When the scoreboard shows 97 points for USA for the first time, output '97'," is modified to "What is the current score for USA?" (See Figure 13) As shown in the right part of Figure 5, the model performance improves significantly. This indicates that the model struggles to adapt to the proactive output format, and further targeted improvements are needed.

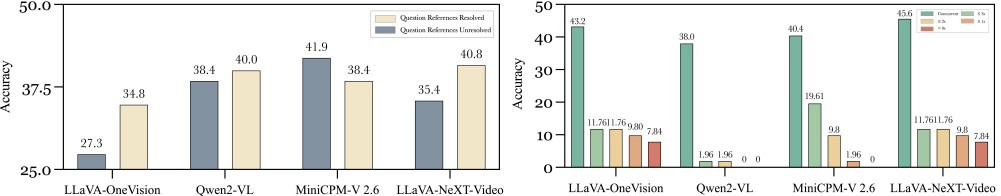

Figure 5: **Left:** Performance comparison of top open-source MLLMs on the SQA task, with or without reference resolution in questions. **Right:** Performance comparison on the PO task, before and after transforming the question into a concurrent type.

## 5    CONCLUSION

In this work, we introduce StreamingBench, the first comprehensive benchmark designed to assess the streaming video understanding capabilities of MLLMs. StreamingBench consists of 900 videos and 4,300 QA pairs, covering 18 tasks across three main categories that evaluate key aspects of streaming video comprehension. Experiments with 13 state-of-the-art MLLMs reveal that even the best-performing model Gemini 1.5 Pro still falls significantly short of human-level performance. Additionally, we analyze the performance gap and propose potential directions for improving MLLMs. We hope that our work will contribute to the development of future AI systems with improved performance in real-world scenarios.

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

# A  MORE DETAILS OF EVALUATION

## A.1  MODEL INFERENCE SETTINGS

**GPT-4o** Limited by API, we extract only 64 frames for each video. In our current environment, more frames will result in a large number of access failures. We will try other methods to use more frames for evaluation in the future.

**Qwen2-VL** To streamline the evaluation process and reduce associated costs, frames are extracted at different rates based on the length of the video: 1 fps for videos shorter than 5 minutes, 0.5 fps for videos between 5 and 10 minutes, and 0.2 fps for videos longer than 10 minutes. When assessing the performance of the four models on ACU tasks with and without high-redundancy visual information inputs (Table 5), we applied a frame extraction strategy similar to that used for videos in order to evaluate multiple images. This approach is more cost-efficient as the processing pipeline for videos incurs lower computation resource consumption per frame compared to standalone images. It is assumed that the model requires fewer computational resources to process a single image when embedded within a video.

**Other Open-Source MLLMs** We adhere to the official inference strategies of these MLLMs. For MiniCPM-V 2.6 and InternVL-V2, we have found that there are some situations where the last few frames cannot be captured. We assume such strategy may affect the evaluation results and plan to solve this in the future.

## A.2  EVALUATION PROTOCALS

Real-time visual understanding tasks, omni-source understanding tasks, ACU and MCU follow the same evaluation process. For each question, we crop the video segment from the full video up to the timestamp where the question appears, and use this segment as the input to the model, while applying the following prompt for multiple-choice question answering:

> **Prompt used for Tasks Except for SQA, PO**
>
> ```
> You are an advanced video question-answering AI assistant.
> You have been provided with some frames from the video and a
> multiple-choice question related to the video.  Your task is
> to carefully analyze the video and provide the best answer to
> question, choosing from the four options provided.  Respond with
> only the letter (A, B, C, or D) of the correct option.
> ```
>
> **Question:** {}
>
> **Options:** {} {} {} {}
>
> **The best option is:**

For the SQA task, we follow a similar protocol to the previous one, with the key difference being that the prompt includes contextual information in textual form. This context consists of the timestamp (as an integer), the questions, answer options, and the ground truth answer from prior conversations. Notably, the prompt provides the ground truth answer instead of the model's previous responses, as we assume that humans can correct incorrect responses during real streaming conversations. During evaluation, the model is presented with a sequence of related questions about the same video, with information from earlier interactions incorporated into the prompt.

---

**Prompt used for SQA**

```
You are an advanced video question-answering AI assistant.  You
have been provided with a video and a multiple-choice question
related to the video.  Your task is to carefully analyze the video
and the provided context to answer the question, choosing from the
four options provided.  Respond with only the letter (A, B, C, or
D) of the correct option.
```

Here are the contextual information related to the video. Please answer the questions based on the contextual information:

At timestamp {}, the following question and answer occurred: Question: {}; Options: {}, {}, {}, {}; Answer: {};
...

Here is the question. Answer it and don't confuse it with the previous conversation.

**Question:** {}

**Options:** {} {} {} {}

**The best option is:**

---

In PO tasks, the questions generally take the form: "When ..., output ...." To enhance the accuracy and stability of the responses, the prompt for PO includes a query about whether an output is necessary. The polling timestamps encompass the query timestamp and every second within the interval [-4,4], using the ground truth timestamp as the origin, up to 10 timestamps.

---

**Prompt used for PO**

```
You are an advanced video question-answering AI assistant.
You have been provided with some frames from the video and a
multiple-choice question related to the video.  Your task is
to carefully analyze the video and provide the best answer to
question, choosing from the four options provided.  Respond with
only the letter (A, B, C, or D) of the correct option.
```

**Question:** {}

Is it the right time to output {}? You can only answer yes or no.

**The answer is:**

---

## B MORE DETAILS OF DATA CONSTRUCTION

### B.1 VIDEO SELECTION

We divide the streaming understanding scenarios into eight distinct categories to ensure a comprehensive simulation of real-world, real-time streaming applications. The Life Record category includes videos that capture everyday activities such as travel vlogs, house tours, and reaction videos. The Competition category features sports, including football, basketball. Video games category includes eSports and gaming videos. Education encompasses videos like lectures, tutorials, and other instructional content. TV Show covers a range of media, including TV series, talk shows, and news segments. Unusual Event focuses on unexpected scenarios such as car accidents, prank videos, and magic shows. The Documentary category features content that includes science documentaries, cultural explorations. Animation & Movie category includes comedies, kid's shows and animated films. This categorization ensures that the benchmark thoroughly simulates the diverse scenarios encountered in real-time streaming environments.

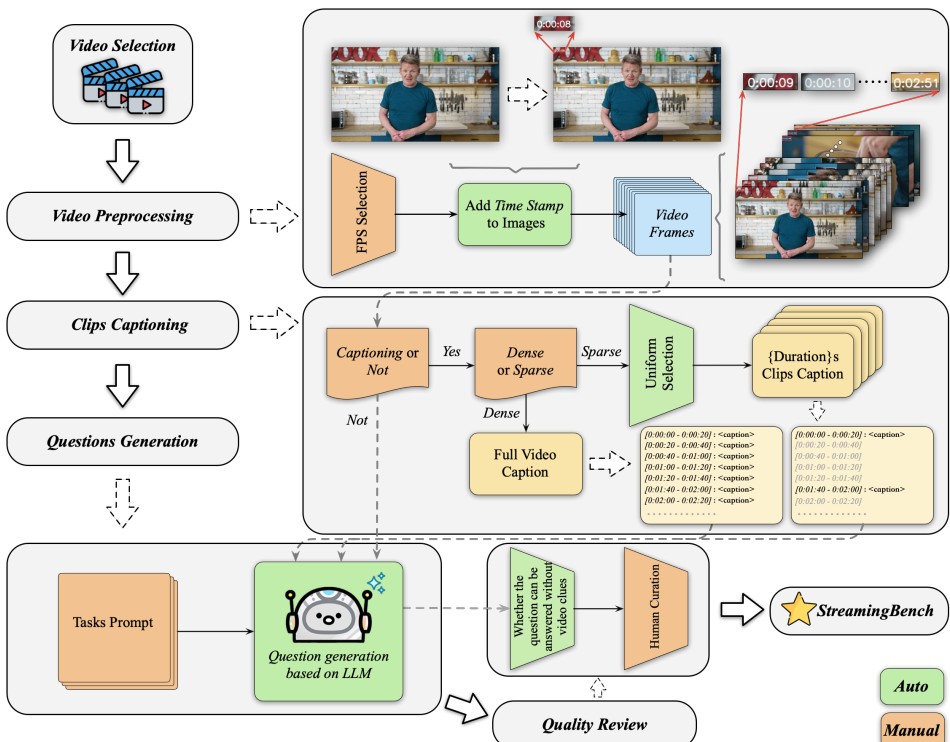

Figure 6: Pipeline of StreamingBench for automatic construction of streaming QA.

## B.2 QA GENERATION

To create questions that truly capture the streaming nature of video understandings, we selected five distinct timestamps for each video to serve as query points. For tasks under the Real-Time Visual Understanding category and the Proactive Output task, we adapted the traditional two-step approach of generating questions based on captions. The pipeline for QA Generation is illustrated in Figure 6. Specifically, we employed GPT-4o to sample frames from the video at a rate of 1 frame per second (fps). We observed that for Single-Frame tasks, directly generating questions based on the sampled images, without an intermediate captioning phase, resulted in higher quality questions. Conversely, for Multi-Frame tasks, generating captions first and then formulating questions from those captions yielded better results. Unlike other video benchmarks where queries are typically presented at the end of the video, StreamingBench introduces queries at various points throughout the video. To automatically generate appropriate query timestamps, we tagged each sampled frame with its corresponding timestamp in the video. We found that this method helped us produce high-quality questions with realistic query timings. Additionally, we tagged each question with the time range during which the relevant clues appeared in the video, specifying the minimal video segment necessary to answer the question accurately. This tagging approach also proved effective, ensuring the generation of high-quality, contextually relevant questions. For tasks in the Omni-Source Understanding category and Contextual Understanding tasks (excluding Proactive Output), where questions require audio information, we employed meticulous manual annotation. Each video was carefully annotated to ensure the precision and relevance of the generated questions.

## B.3 PROMPT FOR QA GENERATION

Below are our prompts for automatically constructing question-answer pairs. First we generate captions, and then generate questions with precise timestamps based on the captions. Alternatively, we can directly generate questions with precise timestamps from images marked with corresponding timestamps

### Prompt used for captions construction

```
You are an AI assistant skilled in video comprehension, captioning,
and adding timestamps.  These are frames from a {} second {SUBJECT}
video with 1-second intervals between each frame.  Each image has a
corresponding timestamp.

Follow these TWO STEPS:

STEP 1: Detailed Description

1.  Describe the video in as much detail as possible, including
features (shapes, sizes, colors, positions, orientations, etc.),
actions, movements, relationships of people and objects, and
backgrounds.
2.  Only describe what is visible in the video.  Do not include
information you are unsure about.
3.  Start the description naturally, without summaries.
4.  Be objective and avoid subjective opinions or guesses.
5.  Write naturally and fluently.  Do not caption frame by frame.
6.  Ensure proper grammar, especially for person and tense.

STEP 2: Add Timestamps

1.  Add specific timestamps to different segments of the
description based on the timestamps in the top left corner of the
frames.
2.  Do not modify the original description content.
3.  Use the format [H:MM:SS – H:MM:SS] for ranges or [H:MM:SS] for
single timestamps.
4.  Ensure timestamps match the corresponding video frames.

Example format:

[H:MM:SS – H:MM:SS]: description segment; [H:MM:SS]: description
segment; ...

Only output the captions with added timestamps.  Do not include
any other content.  Carefully review the provided video frames,
then provide your response.
```

---

**Prompt used for questions generation**

---

You are an AI assistant skilled at generating questions and
answers.  I have a 20s video clips extracted from a {SUBJECT} video,
organized in chronological order with time marks like [0:01:00 –
0:01:20].  the time marks do not start from 00:00:00 if the time
marks is not [0:00:00 – 0:00:20].  Please read the video clips
carefully and provide question-answer pairs based on the video
clips.

Follow these instructions:

**GUIDE:**
1.  Ensure the questions and answers are highly relevant to the
captions and DO NOT INCLUDE TOPICS NOT MENTIONED in the captions.
2.  IGNORE CONTRADICTORY OR UNREASONABLE PARTS of the captions.  Do
not base questions on them.
3.  Present questions as multiple-choice.  Provide task type,
questions, options, and answers.  Each question should have 4
options with similar formats, and the wrong options should be
deceptive.
4.  Avoid questions specific to individual scenes or overly precise
timing.  Consider all scenes as a whole.
6.  Pay attention to grammar.  Avoid grammar mistakes, especially
with person and tense.
7.  Ensure questions are reasonable and challenging, requiring
thoughtful consideration to answer correctly.
8.  The question should not contain phrases like "In the beginning
of the clips" or "at the beginning of the video" or "in the video"
or "in this clips"; it can include expressions of the present or
recent past such as "just now" or "right now."
9.  Please pay attention to the tense of the sentences.
10.  Provide only {NUMBER} best question-answer pairs based on the
caption

**Understand the following task descriptions:**

<Specific Task Definition>

**Example Tasks:**

<Few Shots>

Please generate Q&A content in the following format:
Format:
Task Type:  <task type>
Question:  <question>
Time Stamp:  <time stamp>
A. <option A>
B. <option B>
C. <option C>
D. <option D>
Answer:  <answer>

Output only the questions and answers.  Now, please carefully
review the captions and output your questions and answers following
the SAME FORMAT as the examples above.

## C   MORE EXPERIMENTAL RESULTS

### C.1   RESULTS OF STREAMING VIDEO MLLMS

**Flash-VStream**  Flash-VStream is evaluated on all tasks (except for PO) using the same strategy applied to other models. In most cases, it only produces the output "A", leading to extremely low accuracy on these tasks. For PO, the official scripts are initially attempted; however, due to excessive processing time, we adopt the polling strategy, which yields similarly poor results.

**VideoLLM-online**  We modify the official script to enable evaluation on our benchmark. However, it cannot follow instructions accurately and generate a large amount of redundant information. For PO, VideoLLM-online is able to accept streaming video input, yet the streaming evaluation strategy performs similarly, or even worse, compared to the polling strategy.

The evaluation results are listed in Table 6. We also provide some output examples for both models in the following:

---

**Responses of Flash-VStream**

`Real-time Visual Understanding`

What does the the glass structure likely depict right now?

Answer: "B"

A. A historic monument.
B. A modern bus stop.
C. A construction site.
D. A marketplace.

Flash-VStream: A. A historic monument.

`Proactive Output`

When the dealer reveals the hidden card and it is a face card, output "Dealer card face". Is it the right time to output "Dealer card face"?

Ground truth timestamp: "00:05:44"

Flash-VStream: { "content": "Yes", "time": 336 (00:05:36) }

---

Table 6: Performance of two streaming video MLLMs on StreamingBench.

| Model | Params | Frames | Real-Time Visual Understanding | | | | | | | | | | Omni-Source Understanding | | | | | Contextual Understanding | | | | | Overall |
|---|---|---|---|---|---|---|---|---|---|---|---|---|---|---|---|---|---|---|---|---|---|---|---|
| | | | OP | CR | CS | ATP | EU | TR | PR | SU | ACP | CT | All | ER | SCU | SD | MA | All | ACU | MCU | SQA | PO | All | |
| Human | | | | | | | | | | | Human | | | | | | | | | | | | | |
| Human | - | - | 89.47 | 92.00 | 93.60 | 91.47 | 95.65 | 92.52 | 88.00 | 88.75 | 89.74 | 91.30 | 91.46 | 88.00 | 88.24 | 93.60 | 90.27 | 90.26 | 88.80 | 90.40 | 95.00 | 100 | 93.55 | 91.66 |
| Streaming MLLMs | | | | | | | | | | | | | | | | | | | | | | | | |
| Flash-VStream | 7B | - | 25.89 | 43.57 | 24.91 | 23.87 | 27.33 | 13.08 | 18.52 | 25.20 | 23.87 | 48.70 | 23.23 | 25.91 | 24.90 | 25.60 | 28.4 | 26.00 | 24.80 | 25.20 | 26.80 | 1.96 | 24.12 | 24.04 |
| VideoLLM-online | 8B | - | 39.07 | 40.06 | 34.49 | 31.05 | 45.96 | 32.40 | 31.48 | 34.16 | 42.49 | 27.89 | 35.99 | 31.20 | 26.51 | 24.10 | 32.00 | 28.45 | 24.19 | 29.20 | 30.80 | 3.92 | 26.55 | 32.48 |

---

**Responses of VideoLLM-online**

```
Real-time Visual Understanding
```

What text is prominently shown on the packaging of the darker graphics card right now?

Answer: "C"

A. DESIGNED BY GAMERS, BUILT BY NVIDIA.
B. CREATED BY GAMERS, DEVELOPED BY NVIDIA.
C. INSPIRED BY GAMERS. BUILT BY NVIDIA.
D. ENGINEERED BY GAMERS, MADE BY NVIDIA.

VideoLLM-online: A. DESIGNED BY GAMERS, BUILT BY NVIDIA.

```
Proactive Output
```

When the player in the light blue jersey scores a goal, output G̈oal by light blue team.. Is it the right time to output G̈oal by light blue team?̈

Ground truth timestamp: "00:04:11"

VideoLLM-online: { "content": "You are an advanced video question-answering AI assistant. (Sorry, the last response is wrong) No.assistant: You are an advanced video question-answering", "time": 255 (00:04:15) }

---

Table 7: Performance of various MLLMs on the three core tasks set for streaming understanding capabilities in StreamingBench.

| Model | Video Length | Real-Time Visual Understanding | | | | | | | | | | |
|---|---|---|---|---|---|---|---|---|---|---|---|---|
| | | OR | CR | CS | ATR | EU | TR | PR | SU | ACR | CT | All |
| LLaVA-OneVision | ≤60 s | 84.81 | 75.00 | 84.93 | 91.30 | 89.29 | 85.88 | 82.61 | 73.91 | 73.53 | 63.26 | 81.30 |
| | >60 s | 79.17 | 74.07 | 72.95 | 76.79 | 66.92 | 66.53 | 63.53 | 63.00 | 63.86 | 25.00 | 66.94 |
| Qwen2-VL | ≤60 s | 86.08 | 80.00 | 78.08 | 85.51 | 89.28 | 82.35 | 78.26 | 73.91 | 67.65 | 67.35 | 78.89 |
| | >60 s | 72.22 | 81.18 | 91.30 | 75.11 | 63.91 | 66.95 | 70.59 | 59.50 | 60.00 | 38.89 | 66.33 |
| MiniCPM-V 2.6 | ≤60 s | 88.61 | 75.00 | 83.56 | 89.86 | 75.00 | 81.18 | 82.61 | 69.57 | 77.94 | 79.59 | 81.67 |
| | >60 s | 67.36 | 70.37 | 76.23 | 71.73 | 62.41 | 60.17 | 67.06 | 53.00 | 58.60 | 44.44 | 63.52 |
| Video-LLaMA2 | ≤60 s | 79.22 | 65.00 | 63.01 | 72.46 | 64.29 | 61.18 | 78.26 | 47.83 | 62.69 | 55.32 | 65.06 |
| | >60 s | 49.65 | 53.70 | 55.33 | 54.43 | 52.63 | 37.29 | 29.41 | 41.00 | 41.75 | 17.36 | 44.59 |
| Video-CCAM | ≤60 s | 79.75 | 60.00 | 76.71 | 82.61 | 78.57 | 81.18 | 65.22 | 63.04 | 67.65 | 57.14 | 73.51 |
| | >60 s | 50.0 | 57.41 | 61.48 | 56.54 | 62.41 | 40.25 | 36.48 | 44.00 | 45.26 | 20.83 | 48.26 |
| LongVA | ≤60 s | 82.28 | 70.00 | 61.64 | 79.71 | 78.57 | 71.76 | 78.26 | 60.87 | 64.71 | 57.14 | 70.37 |
| | >60 s | 66.67 | 62.04 | 60.66 | 67.93 | 57.89 | 55.08 | 55.29 | 51.5 | 52.28 | 22.92 | 56.47 |
| Kangaroo | ≤60 s | 83.54 | 75.0 | 76.71 | 85.51 | 78.57 | 77.65 | 73.91 | 65.22 | 76.47 | 8.16 | 71.67 |
| | >60 s | 67.71 | 87.04 | 68.44 | 69.62 | 63.91 | 55.51 | 51.76 | 53.00 | 58.60 | 37.5 | 61.63 |

The complete results regarding the impact of video length on the model's streaming video understanding are presented in Figure 7. The results indicate that the length of the video does indeed affect the model's performance. However, the performance differences in the tasks of causal reasoning and clips summarization are not particularly significant. In contrast, the impact of video length on the model's performance in the counting task is substantial.

## C.2 Details of Human Evaluation

We invited five participants to evaluate the tasks in StreamingBench. For each task, 10% of the questions were randomly selected from StreamingBench and presented to the participants. Each participant had only one chance to respond to each question. Additionally, once a video had been watched, participants were not allowed to rewind or replay it.

# D Data Examples

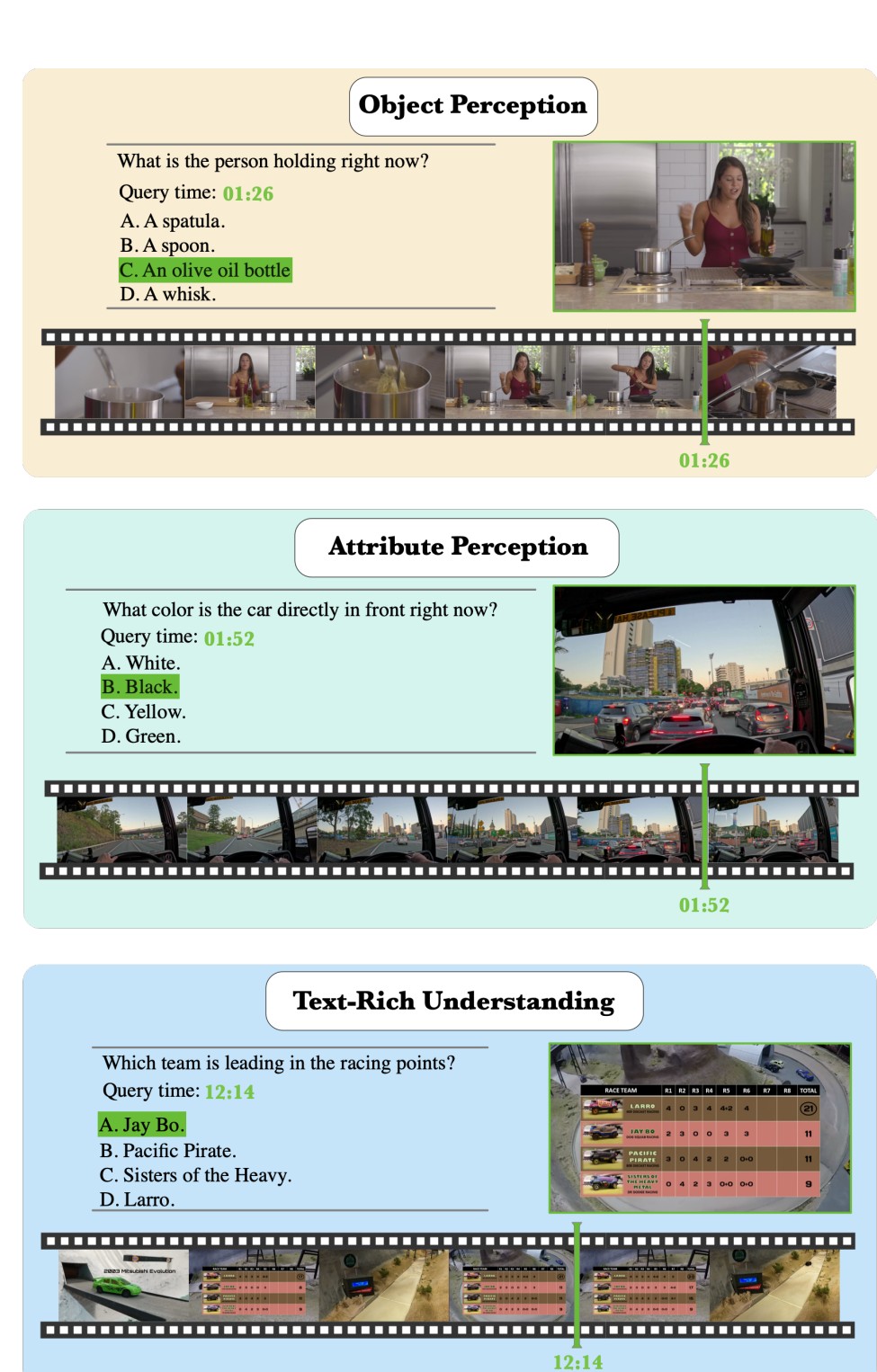

Figure 7: Data examples for object perception, attribute perception, and text-rich understanding tasks.

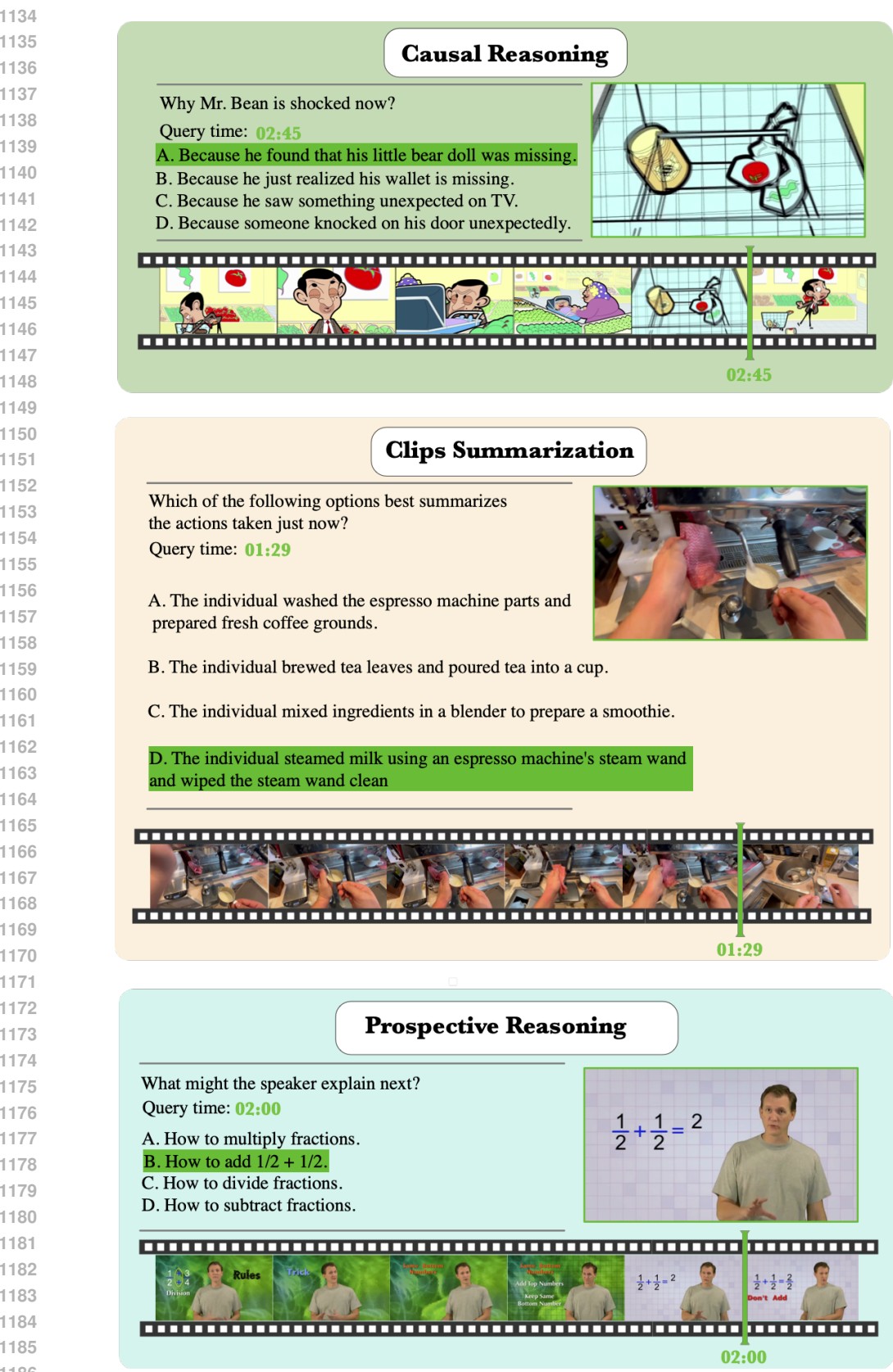

Figure 8: Data examples for causal reasoning, clips summarization , and prospective reasoning tasks.

## Event Understanding

What is seen in the initial scene of the video?

Query time: **00:20**

A. A blue racing car near a curve.
B. Several racing cars are lined up in a row.
C. A green racing car in the pits.
D. A red racing car is parked on the track.

**00:20**

## Emotion Recognition

What is the current emotion of the man with buzz cut of the video, and what caused his emotional changes?

Query time: **01:17**

A. He was very disappointed because the chef said the dishes he made are not tasty.
B. He felt nervous because the chef was tasting the dishes he had made.
C. He felt relieved because the chef did not criticize…
D. He was very happy because the chef praised him should be working in the finest restaurant in the world.

**01:17**

## Scene Understanding

Please describe the scene that just occurred in the video.

Query time: **01:16**

A. A player wearing a black number 8 jersey jumped up and punched. The commentator said, "There's a goal."
B. Five team members dressed in black hugged and congratulated each other. The commentator said, 'From this talented young man.'.
C. A player wearing a black number 8 jersey ran towards the camera. The commentator said, "There's a goal."
D. The No.8 player dressed in black volleyed . The commentator said, "A problem that even Chad Kingston couldn't resist."

**01:16**

Figure 9: Data examples for event understanding, emotion recognition, and scene understanding tasks.

**Source Discrimination**

Who just said 'Look at that little girl over there.'?
Query time: **02:26**

A. a dark-haired male wearing a gray turtleneck sweater…
B. a blonde woman wearing a blue jean jacket with a necklace …
C. a dark-haired male wearing a bright pink short-sleeved …
D. a blonde woman wearing a purple long-sleeved shirt, …

02:26

**Anomaly Context Understanding**

What unusual event just occurred?
Query time: **02:02**

A. The magician took the playing cards chosen by the female guest in his right hand and blew.
B. The black marker in the magician's hand suddenly disappeared.
C. The magician spread a deck of playing cards on the table for the female guest to pick.

D. The white marker in the magician's hand suddenly disappeared.

02:02

**Misleading Context Understanding**

How many red snooker balls are left?
Query time: **02:57**
A. Three red snooker balls remain.
B. There are ten red snooker balls.
C. There are nine red snooker balls.
D. Three red snooker balls remain.

02:57

Figure 10: Data examples for source discrimination, misleading context understanding, and anomaly context understanding tasks.

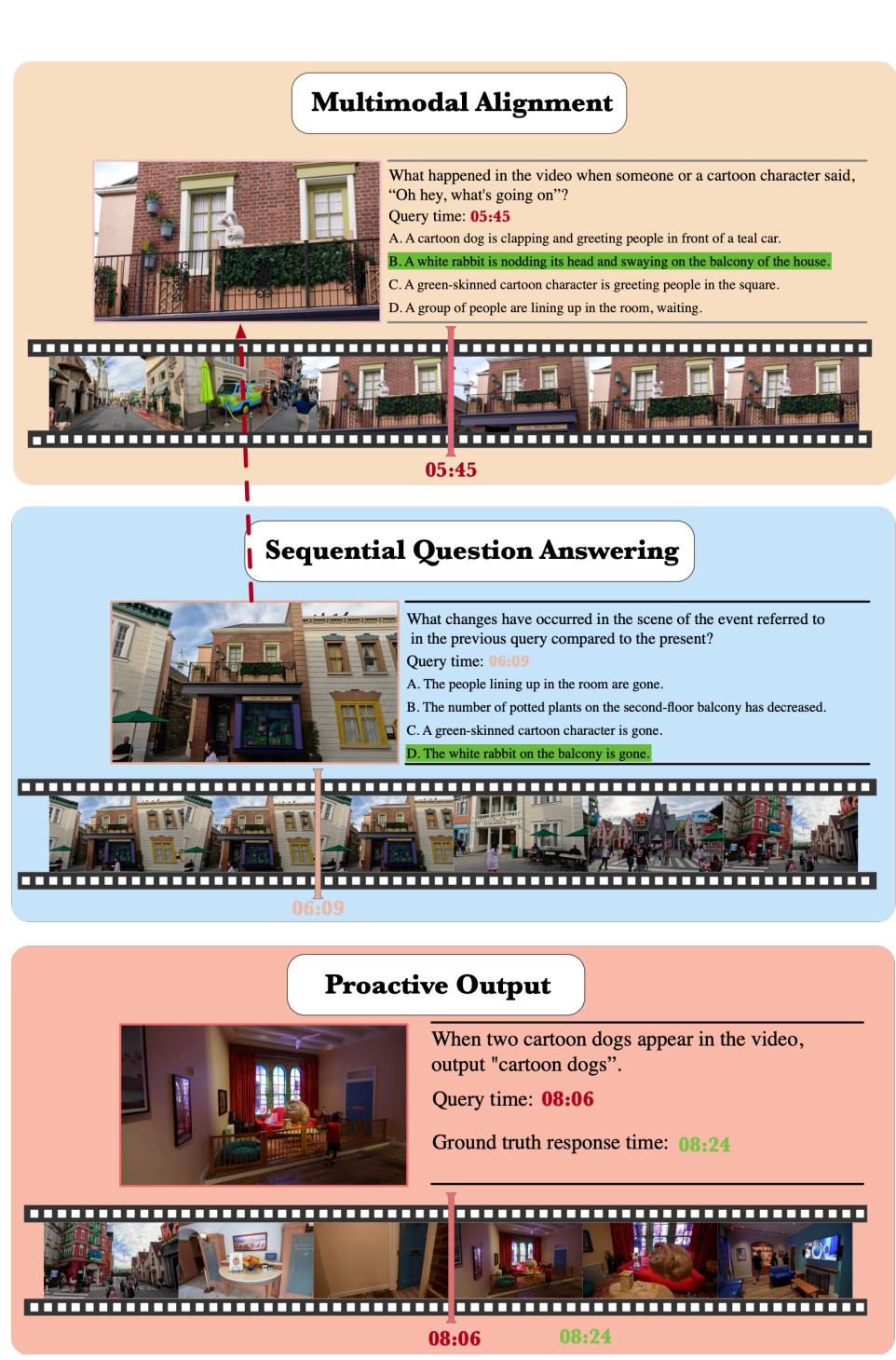

Figure 11: Data examples for multimodal alignment, sequential quension answering, and proactive output tasks.

Figure 12: Data examples for spatial understanding, counting, and action perception tasks.

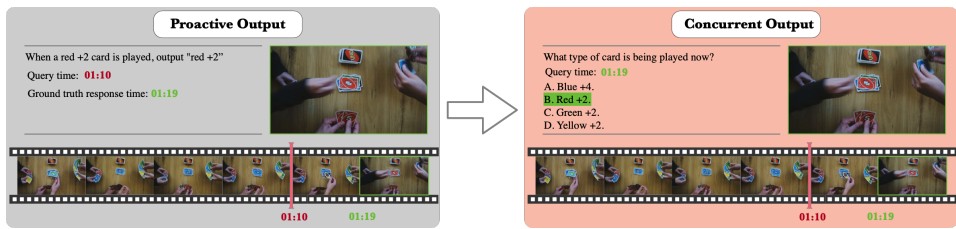

Figure 13: The process of transforming proactive output tasks into a general form concurrent type question.

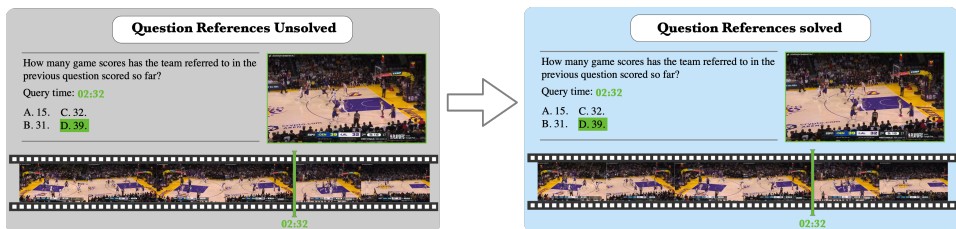

Figure 14: The process of references resolution transformation in sequential question answering.

