# OpenReview forum: "StreamingBench: Assessing the Gap for MLLMs to Achieve Streaming Video Understanding"
_ICLR.cc/2025/Conference — Submitted to ICLR 2025_

### Official Review · Reviewer_Rih2 · 2024-11-03

**Soundness:** 4
**Presentation:** 4
**Contribution:** 3
**Rating:** 8
**Confidence:** 4

**Summary:**

The authors propose a new benchmark, StreamingBench, for evaluating MLLMs in streaming video understanding. It assesses three aspects of streaming video understanding: real-time visual understanding, omni-source understanding, and contextual understanding. There are 18 tasks in total. They evaluate 13 open-source Video MLLMs and 3 proprietary MLLMs on this benchmark and analyze the results.

**Strengths:**

S1: In this paper, the authors propose a benchmark to evaluate the MLLMs' capabilities of streaming video understanding, which is novel and unexplored previously. I believe this will facilitate the advancement of Video MLLMs.

S2: The benchmark considers both video and audio modalities, which have been absent in most previous benchmarks.

S3: The experiments and analysis are comprehensive and detailed, effectively highlighting the limitations of current Video MLLMs in understanding streaming video.

S4: The writing is clear and well-structured.

**Weaknesses:**

W1: The impact of language model size on performance has not been analyzed. For instance, models like InternVL-V2 come in 1B, 2B, 4B, 8B, 26B, 40B, and 72B parameter versions, while Video-LLaMA2, LLaVA-OneVision, and Qwen2-VL also have 72B versions. Expanding your experiments to include these variations and providing a more detailed analysis would enhance your work. Additionally, exploring the number of frames the model can process would offer valuable insights.

W2: Several significant models are missing from the evaluation, such as LongVILA [1], Long-LLaVA [2], and Oryx [3]. Including these would provide a more comprehensive comparison.

[1] https://github.com/NVlabs/VILA/blob/main/LongVILA.md

[2] https://github.com/FreedomIntelligence/LongLLaVA

[3] https://github.com/Oryx-mllm/Oryx

**Questions:**

Q1: Do you plan to extend your benchmark to support additional streaming video understanding tasks, such as dense streaming video captioning or grounding? Currently, it only supports QA.

Q2: Could you provide more details on how audio is utilized in the MLLMs? I am aware that Video-LLaMA2 supports audio input, but what about the other models?

Q3: Have you considered memory constraints?  I believe this is a crucial factor in streaming video understanding. It would be more consistent and fair to have a fixed memory limit applied across all models, as retaining all input frames throughout a task—particularly in real-world applications—may not be feasible. If some models retain all frames, it could lead to an unfair advantage.

---

> ### Author Response · Authors · 2024-11-28
> **Response (Part 1)**
>
> Thank you for your detailed and insightful feedbacks. Below, we respond to each of your points in detail:
>
> **Response to Q1: Extension of the benchmark for additional tasks**
> - Thank you for your valuable suggestions! We recognize that there are also other interesting and important streaming video understanding tasks such as dense streaming video captioning and streaming video grounding. We will explore these tasks in the future and try to add them in the next version of StreamingBench.
>
> **Response to Q2: For models that do not support audio input**
> - In the current Omni-Source Understanding experiments, models that do not support audio input only rely on video information to respond to Omni-Source Understanding tasks. As a comparison, we conducted an additional evaluation using Video + ASR information as input. **The detailed results are shown in General response 1.** The results indicate that while the models demonstrated improved performance on the Omni-Source Understanding tasks with additional ASR information, they still fell significantly short of the performance achieved by Gemini 1.5 pro in omni-source understanding tasks. This highlights the necessity of advancing existing MLLMs to natively support audio input in streaming scenarios.
>
> **Response to Q3: Memory Constraint**
> - In our current experiments, we adhere to the standard settings for each model. While we recognize that memory constraints are indeed crucial, addressing them is not the primary focus of our work at this stage. However, even though we did not enforce a fixed memory limit across all models, the memory usage for most models remains within a manageable range (e.g., for 7B models, it does not exceed the capacity of a single A100 80G GPU).
>
> - Additionally, as a preliminary effort to evaluate the impact of memory usage on model performance, we sampled 100 QA data and analyzed the average number of input tokens processed by several models under standard settings. From the results, it seems that for current video MLLMs, processing more input tokens does not necessarily lead to better performance. The specific results are as follows:
>
>     **Table 1: Average Input Tokens and Model Performance**
>     | Model (w/o ASR)      | Avg. Input Token | All(RTVU)  | All(OSU)   | All(CU)    | Overall   |
>     | -------------------- | ---------------- | ----- | ----- | ----- | ----- |
>     | LLaVA-OneVision-7B   | 6426.91          | 71.12 | 38.40 | 31.63 | 56.16 |
>     | Qwen2-VL-7B          | 56081.45         | 69.04 | 34.90 | 30.37 | 53.91 |
>     | MiniCPM-V 2.6-8B     | 12696.77         | 67.44 | 35.00 | 34.21 | 53.71 |
>     | LLaVA-Next-Video-32B | 5029.79          | 66.96 | 34.90 | 30.04 | 52.64 |
>     | InternVL-8B          | 8599.95          | 63.72 | 35.80 | 30.59 | 51.06 |
>     | Kangaroo             | 8420.45          | 64.60 | 34.20 | 29.32 | 50.97 |
>     | LongVA               | 18585.91         | 59.96 | 35.40 | 29.34 | 48.55 |
>     | VILA-1.5             | 3319.45          | 52.32 | 33.10 | 27.24 | 43.18 |
>     | Video-CCAM           | 1179.99          | 53.96 | 29.70 | 21.83 | 42.34 |
>     | Video-LLaMA2         | 1124.50          | 49.52 | 32.40 | 22.08 | 40.43 |
>
> -  With more video MLLMs emerging for streaming video understanding, we will include memory usage and constraints as a consideration in future work. Thank you again for your suggestion!

---

> ### Author Response · Authors · 2024-11-28
> **Response (Part 2)**
>
> **Response to W1: Impact of model size**
> - Thank you for your suggestion. We conducted additional evaluations on models of various sizes, including LLaVA-OneVision 0.5B & 72B, Qwen2-VL 2B, InternVL-V2 2B & 72B, and VideoLLaMA2 72B. **Overall, stronger LLM backbones proved to be beneficial.** Due to resource and time constraints, we were unable to complete evaluations for some models. We will update the results in the paper once these evaluations are completed.
>
>     **Table 2: Impact of Model Size**
>     | Models                   | OP    | CR    | CS    | ATP   | EU    | TR    | PR    | SU    | ACP   | CT    | All（RTVU) | ER    | SCU   | SD    | MA    | All(OSU)  | ACU   | MCU   | SQA   | PO    | All(CU)   | Overall     |
>     | ------------------------ | ----- | ----- | ----- | ----- | ----- | ----- | ----- | ----- | ----- | ----- | ---------- | ----- | ----- | ----- | ----- | --------- | ----- | ----- | ----- | ----- | --------- | --------- |
>     | **LLaVA-OneVision-0.5B** | 65.12 | 61.72 | 65.3  | 65.36 | 56.52 | 58.57 | 57.41 | 53.25 | 55.52 | 33.16 | **58.28**  | 37.20 | 24.40 | 29.20 | 34.00 | **31.20** | 29.20 | 30.40 | 28.40 | 9.80  | **28.09** | **46.36** |
>     | LLaVA-OneVision-7B       | 80.38 | 72.22 | 76.03 | 80.72 | 68.32 | 71.65 | 67.59 | 65.45 | 65.72 | 45.08 | 71.12      | 40.80 | 37.20 | 33.60 | 44.80 | 38.40     | 35.60 | 36.00 | 27.27 | 11.76 | 31.63     | 56.16     |
>     | **LLaVA-OneVision-72B**  | 79.56 | 85.94 | 82.97 | 83.28 | 78.75 | 74.77 | 81.48 | 67.87 | 73.37 | 51.3  | **75.98**  | 44.80 | 26.80 | 43.60 | 55.20 | **42.60** | 40.40 | 44.80 | 36.40 | 21.57 | **39.33** | **61.39** |
>     | **Qwen2-VL-2B**          | 61.85 | 68.75 | 60.57 | 63.73 | 56.52 | 58.26 | 60.19 | 50    | 50.42 | 37.82 | **56.76**  | 36.40 | 23.20 | 27.20 | 36.40 | **30.80** | 24.40 | 23.20 | 40.80 | 5.88  | **27.96** | **45.36** |
>     | Qwen2-VL-7B              | 75.20 | 82.81 | 73.19 | 77.45 | 68.32 | 71.03 | 72.22 | 61.19 | 61.47 | 46.11 | 69.04      | 41.20 | 22.00 | 32.80 | 43.60 | 34.90     | 31.20 | 26.00 | 39.60 | 1.96  | 30.37     | 53.91     |
>     | **Qwen2-VL-72B**         | 75.48 | 82.03 | 79.18 | 79.41 | 74.53 | 71.03 | 76.85 | 64.63 | 69.69 | 49.22 | **72.28**  | 39.20 | 36.00 | 33.60 | 53.60 | **40.60** | 31.60 | 38.40 | 40.80 | 9.80  | **35.21** | **58.01** |
>     | VideoLLaMA2-7B           | 56.40 | 57.81 | 65.30 | 62.75 | 64.60 | 43.61 | 39.81 | 42.68 | 45.61 | 35.23 | 49.52      | 30.40 | 32.40 | 30.40 | 36.00 | 32.40     | 24.80 | 26.80 | 18.67 | 1.96  | 22.08     | 40.43     |
>     | **VideoLLaMA2-72B**      | 64.85 | 75.78 | 77.29 | 72.79 | 66.25 | 58.88 | 59.26 | 53.66 | 58.07 | 48.19 | **63.70**  | 43.20 | 24.40 | 35.20 | 49.60 | **38.10** | 38.80 | 30.00 | 36.80 | 13.73 | **33.83** | **52.19** |
>     | **InternVL-V2-2B**       | 56.95 | 50.78 | 58.04 | 64.71 | 57.14 | 52.65 | 50.93 | 44.72 | 46.46 | 47.67 | **53.52**  | 32.00 | 30.00 | 29.20 | 43.60 | **33.70** | 32.40 | 26.80 | 37.20 | 13.73 | **30.96** | **44.71** |
>     | InternVL-V2-7B           | 68.12 | 60.94 | 69.40 | 77.12 | 67.70 | 62.93 | 59.26 | 53.25 | 54.96 | 56.48 | 63.72      | 37.60 | 26.40 | 37.20 | 42.00 | 35.80     | 32.00 | 31.20 | 32.32 | 11.76 | 30.59     | 51.06     |
>
>
> **Response to W2: Adding more models for evaluation**
> - Thanks for your suggestions. We have evaluated LongVILA 8B, LongLLaVA 9B, and Oryx 7B. The evaluation results are shown in the following table.
>
>     **Table 3: Results on more models**
>     | Models               | OP    | CR    | CS    | ATP   | EU    | TR    | PR    | SU    | ACP   | CT    | All(RTVU)   | ER     | SCU    | SD     | MA     | All(OSU)    | ACU    | MCU    | SQA    | PO     | All(CU)     | Overall    |
>     | -------------------- | ----- | ----- | ----- | ----- | ----- | ----- | ----- | ----- | ----- | ----- | ------ | ------ | ------ | ------ | ------ | ------ | ------ | ------ | ------ | ------ | ------ | ------ |
>     | LongVILA - 9B       | 67.85 | 49.22 | 60.25 | 71.57 | 56.52 | 56.7  | 40.74 | 56.5  | 57.51 | 41.97 | **58.48** | 37.20 | 25.20 | 30.40 | 40.40 | **33.30** | 29.20 | 28.80 | 25.60 | 7.84  | **26.59** | **46.60** |
>     | Oryx - 7B          | 51.50 | 48.44 | 57.1  | 52.5  | 59.63 | 41.74 | 37.04 | 48.78 | 48.16 | 13.99 | **47.20** | 31.60 | 23.20 | 19.20 | 40.80 | **28.70** | 29.60 | 25.20 | 33.20 | 5.88  | **27.84** | **39.29** |
>     | LongLLaVA - 8B            | 27.32 | 43.75 | 39.87 | 36.39 | 30.63 | 28.35 | 25.92 | 29.8  | 26.35 | 39.38 | **32.18** | 32.40 | 24.80 | 25.60 | 28.00 | **27.70** | 24.00 | 24.40 | 33.20 | 7.84 | **25.97** | **29.98** |

---

> ### Comment · Reviewer_Rih2 · 2024-12-01
>
> Thank you very much for your response. Most of my concerns have been addressed, and your experiments appear more thorough now. I appreciate the effort. However, I’m still curious about your more detailed plans to expand your benchmark to cover a broader range of streaming video understanding, as I genuinely believe this is a highly promising and significant area of research that extends far beyond QA tasks. Of course, there’s no need to provide detailed plans here immediately. Just look forward to seeing your future work.

---

### Official Review · Reviewer_wiNw · 2024-11-03

**Soundness:** 4
**Presentation:** 3
**Contribution:** 3
**Rating:** 6
**Confidence:** 5

**Summary:**

The paper introduces StreamingBench, a benchmark designed to evaluate the streaming video understanding abilities of Multimodal Large Language Models (MLLMs). Traditional MLLMs are effective in offline video comprehension but struggle with real-time, streaming scenarios that require instant processing, synchronizing visual and audio inputs, and understanding context over time. StreamingBench addresses this by presenting 900 videos across diverse real-world scenarios, structured into 18 tasks and 4,300 human-curated question-answer pairs. These tasks test MLLMs on real-time visual, omni-source, and contextual understanding, aiming to bridge the gap between MLLMs and human-level comprehension in streaming contexts. Testing 13 MLLMs, including state-of-the-art proprietary models, revealed significant limitations in current models, especially in omni-source and contextual tasks, suggesting that MLLMs need further development to match human performance in real-time understanding. In general, it is a solid paper and I would recommend acceptance to it.

**Strengths:**

1. It is the first valid benchmark on streaming long videos. The questions are designed properly to reflect the information gained in a streaming long video, and highly resembles what human will ask when continuously watching a video.
2. The evaluation and discussion are both very solid.
3. Human performance is another plus.

**Weaknesses:**

1. The real-time understanding part is nice, but seems a little bit trivial. From all kinds of NIAH evaluations, all models can best answer questions near the ending part of the input, and questions related to "current moments" (which is actually the ending part of input as implemented) might not be so important. Would love to see the understanding on "remembering earlier moments" and the discrepancy from "current moments" for LMMs.

2. The omni-source (visual+audio) part is good. However, how are LMMs without audio abilities evaluated? As these `audio'-related questions seem to be mostly about speeches, do authors plan to interleave text ASR into the model for evaluation? At present, sadly we only see a black-box Gemini-1.5-Pro (for which we do not know how they integrate audio and video) being evaluated with audio.

3. A minor suggestion: the omnisource part of the benchmark is related to "referring reasoning" part of LongVideoBench, an interleaved benchmark for frames and ASR texts, which also needs to judge between concurrent video and audio information in a video. As some other long video benchmarks discussed in Tab 1, please also try to discuss it in the revised paper.

**Questions:**

Please see the weaknesses.

---

> ### Author Response · Authors · 2024-11-28
> **Response to weaknesses**
>
> Thank you for the insightful feedback on our paper. Below, we address each of your concerns point-by-point:
>
> **Response 1: Understanding on "remembering earlier moments."**
>
> - In the Real-Time Visual Understanding (RTVU) tasks, we considered not only "current moments" tasks but also "long dependency" tasks, such as **Counting (CT)** and **Causal Reasoning (CR)**. These tasks require models to capture information over extended time windows, avoiding the possibility of answering questions based solely on local information. To verify this, we conducted evaluations for all models in the main experiment while restricting the context to only the first 60 seconds preceding the query time. The results, shown in the table below, reveal that many models suffered performance degradation on tasks like Counting and Causal Reasoning, while their performance on "current moments" tasks generally improved.
> This indicates a trade-off between focusing on "current moments" and addressing "long dependency" in the current models for streaming video understanding. It also demonstrates that the RTVU tasks in StreamingBench are capable of evaluating both types of capabilities simultaneously. In future versions of StreamingBench, we will continue to optimize towards this direction.
>
> **Table1 : Comparison between experiments retaining only the last 60 seconds of video context before the query time and the main experiment (retaining all context before the query time). Negative values indicate a performance drop caused by limiting the context to the last 60 seconds.**
>
> | Models            | OP    | **CR**     | CS   | ATP   | EU    | TR    | PR    | SU    | ACP   | **CT**     | RTVU  |
> | ----------------- | ----- | ------ | ---- | ----- | ----- | ----- | ----- | ----- | ----- | ------ | ----- |
> | Gemini 1.5 pro    | 4.41  | **-2.53**  | 5.70 | 1.98  | -0.83 | -0.82 | 6.15  | -3.91 | 2.92  | 0.52   | 1.70  |
> | GPT-4o            | 3.55  | **-3.49**  | 2.76 | -2.66 | 5.76  | 1.68  | 8.33  | 8.47  | -3.13 | **-6.13** | 1.26  |
> | Claude 3.5 Sonnet | 1.96  | **-3.57**  | 0.41 | 0.67  | 4.06  | 10.17 | 0.57  | -1.06 | -1.44 | 4.53   | 1.60  |
> | LLaVA-OneVision   | 2.45  | 3.12   | 7.20 | 2.61  | -0.62 | 3.12  | 5.56  | 2.84  | 5.38  | **-3.11** | 3.15  |
> | Qwen2-VL          | 0.55  | **-3.12**  | 3.39 | 1.63  | 6.21  | 4.05  | 1.85  | 4.66  | 3.69  | **-4.14** | 2.11  |
> | MiniCPM-V         | 6.27  | 0.79   | 6.26 | 8.17  | 10.56 | 9.66  | 1.90  | 3.13  | -0.30 | **-6.22** | 4.99  |
> | LLaVA-Next-Video  | 1.91  | 0.78   | 6.88 | 3.90  | 8.08  | 3.43  | 5.55  | 3.25  | -1.14 | **-2.07** | 2.87  |
> | InternVL-V2       | 5.72  | 4.69   | 9.40 | 4.91  | 3.73  | 9.97  | 13.89 | 9.76  | 10.48 | **-13.99** | 6.39  |
> | Kangaroo          | 6.45  | **-10.16** | 5.04 | 1.18  | 3.83  | 1.18  | -3.85 | -5.15 | 1.93  | **-5.70** | 1.16  |
> | LongVA            | 2.99  | 3.13   | 5.26 | 3.92  | 2.49  | 3.43  | -0.92 | 3.25  | 1.99  | 3.10   | 3.15  |
> | VILA-1.5          | 17.44 | 8.59   | 3.70 | 15.36 | 16.49 | 8.73  | -2.78 | 2.44  | 10.20 | 1.03   | 9.22  |
> | Video-CCAM        | -1.90 | 5.47   | 8.43 | 0.98  | -6.84 | -4.05 | 8.34  | -6.10 | -1.14 | **-4.67** | -0.54 |
> | Video-LLaMA2      | 4.09  | 4.69   | 5.56 | 2.29  | 1.86  | 2.50  | 1.86  | 4.07  | 2.55  | **-0.51** | 3.06  |
> | Flash-VStream     | 2.99  | **-15.45** | 0.74 | 2.60  | 4.35  | 10.28 | 6.48  | -0.81 | 3.61  | **-22.79** | 3.35  |
> | Videollm-Online   | -2.36 | 3.69   | 4.74 | 2.94  | 0.47  | 4.05  | 8.33  | 0.62  | -2.78 | **-3.54** | 1.07  |
>
> **Response 2: For models that do not support audio input**
> - In the current Omni-Source Understanding experiments, models that do not support audio input only rely on video information to respond to Omni-Source Understanding tasks. For comparison, we conducted an additional evaluation using Video + ASR information as input. The detailed results are shown in General response 1. In summary, adding ASR information enhances performance but still does not match the capabilities of Gemini 1.5 Pro in Omni-Source Understanding tasks. This highlights the necessity of advancing existing MLLMs to natively support audio input in streaming scenarios.
>
> **Response 3: Discussion with LongVideoBench.**
> - Thank you for your valuable suggestion. The "referring reasoning" part of LongVideoBench is indeed similar to the Omni-Source Understanding tasks in terms of requiring an understanding of concurrent video and audio information. The difference lies in that StreamingBench focuses more on streaming scenarios, whereas LongVideoBench emphasizes long-video scenarios, with differences in the specific types of tasks as well. We will include a discussion on LongVideoBench in the revised version of our paper.

---

> > ### Author Response · Authors · 2024-12-02
> >
> > Dear Reviewer,
> >
> > As the rebuttal period is nearing its conclusion, we hope our responses have adequately addressed your concerns. We look forward to engaging in further discussions and value your feedback.
> >
> > Thank you for your time and consideration.

---

### Official Review · Reviewer_szQS · 2024-11-04

**Soundness:** 3
**Presentation:** 3
**Contribution:** 3
**Rating:** 6
**Confidence:** 5

**Summary:**

This paper introduces StreamingBench, a benchmark designed to evaluate the capabilities of MLLMs in understanding online streaming videos. Key features of StreamingBench include the ability to pose questions at any point during the video, rather than requiring the full video to be viewed first. The benchmark also considers both visual and audio inputs, and it takes into account the influence of historical interactions in multi-turn dialogues.

**Strengths:**

- StreamingBench addresses a relatively unexplored area in MLLM research—real-time video understanding. By allowing questions to be asked at any moment and incorporating both audio and visual data, it expands the scope of existing benchmarks.

**Weaknesses:**

- The methodology for collecting 900 videos from YouTube lacks sufficient detail.
- Given that the study focuses on a model's capability that is seldom addressed—real-time video understanding—it would be beneficial to create or curate a specific supervised fine-tuning (SFT) dataset. This would allow for an evaluation of model performance post-SFT.
- There is a lack of exploration into the model’s ability to generate proactive outputs. Designing a corresponding SFT dataset to assess whether the model performs better with prior exposure to similar outputs would provide valuable insights.
- Clarification is needed on how open-source models like Qwen2-VL tackle omni-source understanding problems, particularly in the absence of audio inputs. This comparison could shed light on the robustness of the proposed benchmark.

**Questions:**

See Weaknesses

---

> ### Author Response · Authors · 2024-11-28
> **Response**
>
> Thank you for your insightful feedback. Below, we address each of your points in detail:
>
> **Response 1: Methodology for collecting 900 videos from YouTube**
> - Thank you for pointing this out. Briefly, we selected videos based on the predefined task definitions and characteristics of 18 tasks. When selecting videos, we conducted extensive searches on YouTube to find videos suitable for annotating these tasks according to specific attributes and requirements. For instance, when looking for videos suitable for the Proactive Output task, we required the videos to contain abundant explicit condition-triggering information. When searching for videos suitable for the Source Discrimination task, we needed videos featuring multiple people, where more than one person spoke before the question was asked. For videos suitable for the Misleading Contextual Understanding task, we required the videos to focus on a specific scene or event, with the scene and event undergoing highly dynamic changes throughout the video. Additionally, to ensure diversity in video content, we selected videos from various fields and perspectives. To find these 900 videos, we approximately reviewed a total of 6,000 videos, with a detailed screening ratio of about 15%. We will supplement a detailed explanation in the appendix in the revised version of the paper.
>
> **Response 2: For SFT dataset design**
> - Thank you for your valuable suggestions. We fully agree that constructing an SFT dataset specifically designed for streaming video understanding (including the Real-Time Video Understanding and Proactive Output tasks as you mentioned) is highly important. However, the primary focus of this work remains on evaluating the streaming understanding capabilities of models. In future work, we will make efforts to develop such an SFT dataset.
>
> **Response 3: For models that do not support audio input**
> - In the current Omni-Source Understanding experiments, models that do not support audio input only rely on video information to respond to Omni-Source Understanding tasks. As a comparison, we conducted an additional evaluation using Video + ASR information as input. **Please refer to the General response 1 for details.** The results show that while adding ASR information improves the performance of these models on Omni-Source Understanding tasks, it remains significantly lower than that of Gemini 1.5 Pro. This highlights the importance of advancing current MLLMs to natively support audio input in order to meet the demands of omni-source understanding capabilities for understanding streaming videos.

---

> > ### Author Response · Authors · 2024-12-02
> >
> > Dear Reviewer,
> >
> > As the rebuttal period is nearing its conclusion, we hope our responses have adequately addressed your concerns. We look forward to engaging in further discussions and value your feedback.
> >
> > Thank you for your time and consideration.

---

### Official Review · Reviewer_7TRD · 2024-11-05

**Soundness:** 2
**Presentation:** 3
**Contribution:** 2
**Rating:** 3
**Confidence:** 4

**Summary:**

This work proposes a benchmark called StreamingBench to evaluate video LLM capabilities in streaming settings. StreamingBench introduces several tasks tailored to streaming scenarios, including real-time visual understanding, omni-source understanding, and contextual understanding.

**Strengths:**

- This work introduces a new benchmark designed to evaluate video models in streaming scenarios.
- It conducts insightful experiments, such as "Does Redundant Information Affect Contextual Understanding?", which provide valuable perspectives in this area.

**Weaknesses:**

- Although this benchmark focuses on streaming scenarios, a standard video LLM can handle it effectively with simple preprocessing. For instance, whenever a question arises, the model can process all frames up to that timestamp. With this approach, the benchmark may not differ significantly from traditional video benchmarks. Therefore, it is essential for this benchmark to identify scenarios that cannot be simplified to an offline setting.

- While handling redundant information is indeed critical for video LLMs, this challenge is not exclusive to streaming scenarios; it is a general issue for any long-video task. As a result, the insights from this paper may be overshadowed by findings from benchmarks specifically focused on long-video understanding.

- The annotation process lacks clarity. Specifically, how do human annotators manually label QA pairs for omni-source understanding and other contextual understanding tasks? What measures are in place to ensure the quality of each question, and what specific strategies were employed?

**Questions:**

As mentioned in the weakness

---

> ### Author Response · Authors · 2024-11-26
> **Response (Part 1)**
>
> **Response 1: Difference between StreamingBench and offline video benchmarks**
>
> - StreamingBench is designed to evaluate the streaming understanding capabilities for all video models, not limited to streaming video models exclusively. Existing standard video LLMs can indeed be evaluated on StreamingBench through certain processing methods, which can also be seen as a way to simulate streaming models with standard video LLMs. As described in Section 4.1, this is how we evaluate the majority of current video LLMs.
>
> - We emphasize that the primary distinction between StreamingBench and existing offline benchmarks is the capabilities being assessed. Existing offline benchmarks cannot comprehensively evaluate the capabilities of the models in streaming scenarios, which is the core design principle of StreamingBench. Specifically, this includes:
>
>     - Traditional offline benchmarks typically pose questions about entire videos, lacking the focus on real-time understanding. In contrast, StreamingBench incorporates tasks designed for **Real-Time Visual Understanding** questions like, *"What words are currently shown?"* Here, answers depend on the exact moment of the query, a crucial aspect that offline benchmarks overlook.
>     - Offline benchmarks often lack consideration for synchronized visual and audio inputs in streaming contexts. When answering such questions, models must consider their synchronization, such as *"What is happening in the video when [sound] is made?"*. The **Omni-Source Understanding** tasks in StreamingBench are tailored to this.
>     - Offline benchmarks typically fail to evaluate the contextual interaction and understanding capabilities in streaming scenarios. StreamingBench includes dedicated tasks for this purpose. For instance, in the **Proactive Output (PO)** task, where a model must proactively respond at the precise moment a condition is met (e.g., *"When the red diamond nine appears on the table for the first time, output 'Red diamond nine.'"*). Similarly, the **Sequential Question Answering (SQA)** task requires models to integrate interaction history with video content to answer questions like, *"What actions did the two individuals mentioned in the previous question just take?"* These tasks go beyond the static Q&A format of offline benchmarks.
>
> - In terms of evaluation methods, the first two types of tasks can be directly assessed for standard video LLMs by feeding all video content up to the question's timestamp. For the third type of tasks, however, evaluation can only be conducted through **certain compromised adaptations** that standard offline models can handle. For example, in the SQA task, the context and Q&A are provided in textual form (rather than interleaved with the corresponding video segments). Similarly, for the PO task, we rely on polling the model to determine whether a response is needed, rather than allowing the offline model to autonomously identify the appropriate moment to respond, as these models lack such capabilities.
>
> - In terms of assessed capabilities, StreamingBench provides a comprehensive evaluation of streaming video understanding, offering significant values compared to existing benchmarks.
>
> **Response 2: Compared to redundant information in long-video benchmarks**
>
> - In long video benchmarks, redundant information primarily arises from the excessive length of the videos. In contrast, in StreamingBench, redundant information does not stem from video length but rather from the nature of streaming scenarios. The tasks associated with this aspect are Misleading Context Understanding (MCU) and Anomaly Context Understanding (ACU):
>
>     - The MCU task focuses on the interference caused by repetitive scenes in answering questions, where redundant information originates from repeated scenes. For example, in a card game video, answering the question *"How many cards are on the table now?"* can be challenging because the camera consistently captures the table from a fixed angle, leading to repetitive footage that affects the ability to answer accurately. This is common in streaming scenarios such as video surveillance
>     - The ACU task, on the other hand, emphasizes the identification of anomalous events in everyday scenes, where redundant information refers to all content other than the anomalous event. This type of redundancy is commonly encountered in streaming video scenarios, such as recordings of daily life activities.
>
> - By analyzing these tasks, we aim to highlight the types of redundant information more relevant to streaming scenarios and provide insights into how these factors influence models as well as directions for their future development.

---

> > ### Author Response · Authors · 2024-11-26
> > **Response (Part 2)**
> >
> > **Response 3: Clarification of the annotation process**
> >
> >
> > ### Annotation Process
> >
> > - Annotators are first instructed to familiarize themselves with the entire video content before watching it in a streaming manner. Based on the requirements of each task type, they record timestamps at appropriate points in the video, formulate questions, and create corresponding options and answers. Each video is pre-assigned a task type based on its content, and annotators are required to generate five QA pairs per video.
> >
> > ### Quality Control
> >
> > - Detailed annotation guidelines are provided for each task type, including task definitions, tips, common errors, and reference examples. Annotators must study these documents and complete trial annotations on several videos. During this trial phase, annotators can provide feedback if they encounter uncertainties. The trial annotations are reviewed, and only those who pass the review are allowed to proceed to the formal annotation phase.
> >
> > - Each annotated sample undergoes a quality review by other annotators. The core principles of the review include, but are not limited to:
> >
> >     - Questions must be based solely on video content up to the specified timestamp.
> >     - Questions must align with the requirements of the assigned task type (e.g., for omni-source understanding tasks, questions must require integrating both visual and     audio content for answering).
> >     - The options must be designed to be distractive, meaning that incorrect options should reference scenes that also appear in the video if possible.
> >
> >   The samples that do not meet these requirements, as well as samples with unmatched or incorrect answers, are discarded to ensure quality.
> >
> > - For each of the 18 tasks, we randomly sampled 10% of the questions, which were then independently answered by five volunteers. The overall average accuracy achieved by the volunteers across all 18 tasks was calculated to be **91.66%**.
> >   We also calculated the consistency of different human subjects answering the same question using the formula as:
> >
> >   $$
> >   C = \frac{1}{Q}   \sum_{i=1}^{Q} \frac{\max(X_i)}{n}
> >   $$
> >
> >   where
> >   $C$ is the overall consistency score.
> >   $Q$ is the total number of questions.
> >   $X_i$ represents the frequency distribution of responses for the $i$-th question.
> >   $\max(X_i)$ is the count of the most frequently chosen option for the $i$-th question.
> >   $n$ is the total number of respondents.
> >
> >   The overall consistency score of **91.95%**, highlights the high quality of the data in StreamingBench.

---

> > > ### Comment · Reviewer_7TRD · 2024-11-26
> > >
> > > If this paper aims to evaluate the capabilities of streaming models, the tasks it designs should specifically target streaming scenarios and emphasize the distinct advantages of streaming models. However, many of the tasks listed in the paper are not exclusive to streaming models. For instance, tasks like object perception and attribute perception can also be effectively handled by offline models with minimal pre-process. This is also reflected in Table 2 of the paper.
> > >
> > > Moreover, if many of these tasks can be reformulated for offline models to handle, it implies that the capabilities measured by these tasks can already be well-assessed by existing offline benchmarks. This raises the question: why do we need another benchmark to evaluate such tasks? For example, tasks like object recognition and attribute perception are already thoroughly evaluated by existing offline benchmarks, making it unnecessary to introduce a new one for the same purpose.
> > >
> > > That said, I do appreciate the design of the “Proactive Output” task, as it effectively demonstrates the unique strengths of streaming models. Compared to offline models, streaming models indeed excel in real-time reasoning and prediction. However, this task includes only 50 examples, which I believe is insufficient to establish a robust benchmark.
> > >
> > > In conclusion, I recommend that this paper focus more on tasks that offline models cannot achieve or struggle to handle. By designing a richer set of tasks that showcase the unique strengths of streaming models, it can better highlight the value of this new benchmark.

---

> > > > ### Author Response · Authors · 2024-11-28
> > > >
> > > > We understand your concern that, after some form of conversion, the first category of Real-Time Visual Understanding (RTVU) tasks may appear similar to tasks in previous offline video benchmarks. This raises the possibility that the value of our benchmark might already be reflected in earlier offline benchmarks. However, we would like to defend the value of StreamingBench from the following perspectives:
> > > >
> > > > 1. We would like to emphasize that the correct evaluation method for StreamingBench is as follows: **video models should process video streams directly, answering questions at different timestamps while maintaining the video stream input**. Consequently, the evaluation requires streaming tasks, where multiple questions are posed at different timestamps within the same video stream. In contrast, existing offline video benchmarks lack streaming tasks, as they only pose questions at the end of the entire video. This makes StreamingBench well-suited for supporting the development and evaluation of future streaming models.
> > > >
> > > > 2. Furthermore, Our evaluation framework is not confined to streaming video models alone. We also want to understand how existing offline models perform in different streaming scenarios. Although these models are not natively designed for streaming evaluation, we adapted the evaluation process to make this assessment possible. However, this adaptation does not compromise the intrinsic streaming nature of the tasks. The tasks are specifically designed for "real-time" Q&A in streaming scenarios and answers may vary over time as the video progresses. Such considerations have not been explicitly addressed in previous offline benchmarks.
> > > >
> > > > 3. The unique characteristics of RTVU tasks make their evaluation essential for several reasons:
> > > >
> > > >     (1) Since prior benchmarks did not explicitly focus on such problems, the performance of current models on these tasks remains unclear. Even if the evaluation reveals that existing offline models perform well on these tasks, the resulting insights and evaluation process hold significant value.
> > > >
> > > >     (2) Please note that despite high accuracy in this adapted evaluation setting, offline models face limitations in speed, computational cost, and context length, making them unsuitable for handling streaming data effectively. Therefore, this evaluation should be interpreted as evidence of the model's potential in understanding streaming content, but it still requires modifications to adapt to streaming scenarios.
> > > >
> > > >     (3) Given that current models are predominantly offline, their performance may deteriorate when adapted into streaming models. Our evaluation benchmark serves as both a reference point and a means to monitor such performance shifts.
> > > >
> > > >     (4) In streaming video understanding, additional dimensions other than only answer accuracy (such as reasoning speed) will become critical in the future. Our benchmark, by virtue of its streaming nature, lays the foundation for the future evaluations.
> > > >
> > > > 4. In addition, StreamingBench already includes several tasks that offline models struggle with. The Omni-Source Understanding task, for example, primarily assesses the ability to synchronize audio and video, which is one of the core capabilities of streaming models. Existing offline video benchmarks have minimal focus on this ability. StreamingBench also includes tasks like SQA and PO, which are exclusive to streaming models. Note that PO actually contains 250 items, but since offline video models lack the ability to perform such tasks, we adopted a polling method for evaluation, which is highly resource-intensive (The actual cost is that each question is asked at least 10 times on average, and each time it is asked, different video inputs are used). Therefore, we only used 50 items when evaluating the existing models.
> > > >
> > > > We also acknowledge your suggestion that as streaming models continue to evolve, future focus should shift toward tasks that streaming models can perform but offline models cannot. This is precisely the direction we aim to develop in the future. However, we believe that all current components of StreamingBench are valuable for evaluating streaming video understanding capabilities and for developing models in this domain. We hope you can reassess our work and provide a fair evaluation.

---

> > > > > ### Author Response · Authors · 2024-12-02
> > > > >
> > > > > Dear Reviewer,
> > > > >
> > > > > As the rebuttal period is nearing its conclusion, we hope our responses have adequately addressed your concerns. We look forward to engaging in further discussions and value your feedback.
> > > > >
> > > > > Thank you for your time and consideration.

---

### Author Response · Authors · 2024-11-28
**General response to all reviewers**

We appreciate that most reviewers acknowledge the value of StreamingBench as the first benchmark to evaluate streaming video understanding capabilities, which are underexplored by previous video benchmarks. In this response, we address the common questions raised by the reviewers. Detailed replies to individual comments from each reviewer are provided within their respective reviews.

**General response 1: About audio input for Omni-Source Understanding tasks**

In the current Omni-Source Understanding experiments in the paper, we did not perform additional processing for models that do not support audio input; these models could only rely on video information to answer Omni-Source Understanding tasks.
We also evaluate all open-source MLLMs across all tasks using a combination of video and ASR information, with detailed results presented in the table below. The results show that while the performance of these models on Omni-Source Understanding tasks has improved, it remains significantly lower than that of Gemini 1.5 Pro. This highlights the critical importance of advancing current MLLMs to natively support audio input in order to meet the demands of omni-source understanding capabilities for understanding streaming videos.

**Table: Model performance with and without ASR input information from videos.**

| Model                        | Params | Frames      | All (RTVU) | All (OSU) | All (CU) | Overall |
|------------------------------|--------|-------------|------------|-----------|----------|---------|
| Gemini 1.5 Pro | -   | Video | 75.69      | 60.22     | 47.79   | 66.90   |
| LLaVA-OneVision (w/ ASR)     | 7B     | 32          | 73.47      | 41.50     | 34.58    | 58.79   |
| LLaVA-OneVision (w/o ASR)    | 7B     | 32          | 71.12      | 38.40     | 32.74    | 56.36   |
| LLaVA-NeXT-Video (w/ ASR)    | 32B    | 64          | 70.27      | 43.20     | 35.46    | 57.49   |
| LLaVA-NeXT-Video (w/o ASR)   | 32B    | 64          | 66.96      | 34.90     | 30.79    | 52.77   |
| Qwen2-VL (w/ ASR)            | 7B     | 0.2-1 fps†  | 70.68      | 41.20     | 36.33    | 57.43   |
| Qwen2-VL (w/o ASR)           | 7B     | 0.2-1 fps†  | 69.04      | 34.90     | 31.66    | 54.14   |
| MiniCPM-V 2.6 (w/ ASR)       | 8B     | 32          | 67.94      | 39.60     | 38.08    | 55.79   |
| MiniCPM-V 2.6 (w/o ASR)      | 8B     | 32          | 67.44      | 35.00     | 34.97    | 53.85   |
| InternVL-V2 (w/ ASR)         | 8B     | 16          | 68.30      | 37.60     | 33.58    | 54.70   |
| InternVL-V2 (w/o ASR)        | 8B     | 16          | 63.72      | 35.80     | 32.42    | 51.40   |
| LongVA (w/ ASR)              | 7B     | 128         | 63.77      | 37.60     | 34.46    | 52.23   |
| LongVA (w/o ASR)             | 7B     | 128         | 59.96      | 35.40     | 29.95    | 48.66   |
| Kangaroo (w/ ASR)            | 7B     | 64          | 65.85      | 31.90     | 28.84    | 51.06   |
| Kangaroo (w/o ASR)           | 7B     | 64          | 64.60      | 34.20     | 30.06    | 51.10   |
| Video-CCAM (w/ ASR)          | 14B    | 96          | 57.84      | 35.90     | 27.72    | 47.13   |
| Video-CCAM (w/o ASR)         | 14B    | 96          | 53.96      | 29.70     | 22.88    | 42.53   |
| Video-LLaMA2 (w/ ASR)        | 7B     | 32          | 53.19      | 30.40     | 26.72    | 42.96   |
| Video-LLaMA2 (w/o ASR)       | 7B     | 32          | 49.52      | 32.40     | 21.93    | 40.40   |
| VILA-1.5 (w/ ASR)            | 8B     | 14          | 50.61      | 24.00     | 24.34    | 39.53   |
| VILA-1.5 (w/o ASR)           | 8B     | 14          | 52.32      | 33.10     | 27.35    | 43.20   |

---

### Meta-Review · Area_Chair_JNmB · 2024-12-21

**Metareview:**

The paper presented a benchmark for stream video understanding. The paper received mixed ratings from four reviewers. Although some of the reviewers appreciated the importance of stream video understanding and the creation of a corresponding benchmark to advance this research area, there are some concerns still remaining in the current paper. First, as pointed out by reviewer 7TRD, the paper needs more discussion on particular insights of real-time stream video understanding, which is considered a different direction from the existing long-video understanding in an offline setting, including from both the benchmark, the method, and the evaluation levels. Both reviewers szQS and wiNw also mentioned in the weakness part that the unique capability of the model for handling online real-time stream video understanding is seldom addressed. Second, the data processing details are not clearly presented in the paper, and some additional analysis experiments should be added to better show the effectiveness of the model. Based on these significant comments, AC finally decided to reject the submission for this time.

**Additional Comments On Reviewer Discussion:**

The reviewers are not fully satisfied with several key points, including the particular focus, designs, and insights on the real-time online video understanding task, the details of the data processing, and some analysis experiments of the model.

---

### Decision · Program_Chairs · 2025-01-22

Reject